



# Impact of dust enrichment on Mediterranean plankton communities under present and future conditions of pH and temperature: an experimental overview

Frédéric Gazeau[1], Céline Ridame[2], France Van Wambeke[3], Samir Alliouane[1], Christian Stolpe[1],
Jean-Olivier Irisson[1], Sophie Marro[1], Jean-Michel Grisoni[4], Guillaume De Liège[4], Sandra Nunige[3],
Kahina Djaoudi[3], Elvira Pulido-Villena[3], Julie Dinasquet[5,6], Ingrid Obernosterer[6], Philippe Catala[6],
Cécile Guieu[1]
[1] Sorbonne Université, CNRS, Laboratoire d'Océanographie de Villefranche, LOV, 06230
Villefranche-sur-Mer, France
[2] CNRS-INSU/IRD/MNHN/UPMC, LOCEAN: Laboratoire d'Océanographie et du Climat:
Expérimentation et Approches Numériques, UMR 7159, 75252 Paris Cedex 05, France
[3] Aix-Marseille Université, Université de Toulon, CNRS/INSU, IRD, MIO, UM 110, 13288,
Marseille, France
[4] Sorbonne Université, CNRS, Institut de la Mer de Villefranche, IMEV, 06230
Villefranche-sur-Mer, France
[5] Scripps Institution of Oceanography, University of California San Diego, USA
[6] CNRS, Sorbonne Université, Laboratoire d'Océanographie Microbienne, LOMIC, F-66650
Banyuls-sur-Mer, France
Correspondence to: Frédéric Gazeau (f.gazeau@obs-vlfr.fr)
Keywords: Mediterranean Sea; Atmospheric deposition; Plankton community ; Ocean acidification;
Ocean warming





# Abstract

In Low Nutrient Low Chlorophyll areas, such as the Mediterranean Sea, atmospheric fluxes represent a considerable external source of nutrients likely supporting primary production especially during stratification periods. These areas are expected to expand in the future due to lower nutrient supply from sub-surface waters caused by enhanced stratification, likely further increasing the role of atmospheric deposition as a source of new nutrients to surface waters. Yet, whether plankton communities will react differently to dust deposition in a warmer and acidified environment remains an open question. The impact of dust deposition both in present and future climate conditions was assessed through three perturbation experiments in the open Mediterranean Sea. Climate reactors (300 L) were filled with surface water collected in the Tyrrhenian Sea, Ionian Sea and in the Algerian basin during a cruise conducted in May/June 2017 in the frame of the PEACETIME project. The experimental protocol comprised two unmodified control tanks, two tanks enriched with a Saharan dust analog and two tanks enriched with the dust analog and maintained under warmer (+3 ºC) and acidified (-0.3 pH unit) conditions. Samples for the analysis of an extensive number of biogeochemical parameters and processes were taken over the duration of the experiments (3-4 d). Here, we present the general setup of the experiments and the impacts of dust seeding and/or future climate change scenario on nutrients and biological stocks. Dust addition led to a rapid and maximum input of nitrate whereas phosphate release from the dust analog was much smaller. Our results showed that the impacts of Saharan dust deposition in three different basins of the open Northwestern Mediterranean Sea are at least as strong as those observed previously in coastal waters. However, interestingly, the effects of dust deposition on biological stocks were highly different between the three investigated stations and could not be attributed to differences in their degree of oligotrophy but rather to the initial metabolic state of the community.



Finally, ocean acidification and warming did not drastically modify the composition of the

autotrophic assemblage with all groups positively impacted by warming and acidification,

suggesting an exacerbation of effects from atmospheric dust deposition in the future.



## 1. Introduction

Atmospheric deposition is well recognized as a significant source of micro- and macro-nutrients for surface waters of the global ocean (Duce et al., 1991; Jickells et al., 2005; Moore et al., 2013). The potential modulation of the biological carbon pump efficiency and the associated export of carbon by atmospheric deposition events are still poorly understood and quantified (Law et al., 2013). This is especially true for Low Nutrient Low Chlorophyll (LNLC) areas where atmospheric fluxes can play a considerable role in nutrient cycling and that represent 60% of the global ocean surface area (Longhurst et al., 1995) as well as 50% of global carbon export (Emerson et al., 1997). These regions are characterized by a low availability of macronutrients (N, P) and/or metal micronutrients (e.g. Fe) that can severely limit or co-limit phytoplankton growth during large periods of year. The Mediterranean Sea is a perfect example of these LNLC regions and exhibits chlorophyll *a* concentrations of less than 0.2 µg L$^{-1}$ all year round over most of its area, except in the Ligurian Sea where relatively large blooms can be observed in late winter-early spring (e.g. Mayot et al., 2016). Recent assessments showed that the atmospheric input of nutrients in the Mediterranean Sea is of the same order of magnitude as riverine inputs (Powley et al., 2017), making the atmosphere a considerable external source of nutrients (Richon et al., 2018). Atmospheric depositions are mostly in the form of pulsed inputs of aerosols from both natural (Saharan dust) and anthropogenic origins (e.g. Bergametti et al., 1989; Desboeufs et al., 2018). Dust deposition is mainly associated with wet deposition and occurs in the form of extreme events (Loÿe-Pilot and Martin, 1996). Ternon et al. (2010) reported on an average annual dust flux over four years of 11.4 g m$^{-2}$ yr$^{-1}$ (average during the period 2003–2007) at the DYFAMED station in the Northwestern Mediterranean Sea. In this region, the most important events reported in the 2010 decade amounted to ~22 g m$^{-2}$ (Bonnet and Guieu, 2006; Guieu et al., 2010b). Atmospheric





deposition provides new nutrients to the surface waters (Guieu et al., 2010b; Kouvarakis et al.,
2001; Markaki et al., 2003; Ridame and Guieu, 2002), Fe (Bonnet and Guieu, 2006) and other trace
metals (Desboeufs et al., 2018; Guieu et al., 2010b; Theodosi et al., 2010), that represent significant
inputs likely supporting the primary production especially during the stratification period (Ridame
and Guieu 2002, Bonnet et al. 2005), although no clear correlation between dust and ocean color
could be evidenced from long series of satellite observation (Guieu and Ridame, 2020).
Experimental approaches have shown that wet dust deposition events in the Northwestern
Mediterranean Sea (the dominant deposition mode in that basin) present a highest positive impact,
by supplying bioavailable new nutrients, compared to dry deposition on all tested parameters and
processes (Guieu et al., 2014a), except for $N_2$ fixation (Ridame et al., 2014). This so-called
fertilizing effect has been experimentally shown using microcosms or mesocosms where the wet
deposition of Saharan dust analog strongly stimulated primary production and phytoplankton
biomass (Guieu et al., 2014a; Ridame et al., 2014) but also modified the phytoplankton diversity
(Giovagnetti et al., 2013; Lekunberri et al., 2010; Romero et al., 2011). However, besides
phytoplankton, dust deposition modified also the bacterial community assemblage and led to even
stronger enhancements of production and/or respiration rates (Pulido-Villena et al., 2014). The
budgets established from four artificial seeding experiments during the DUNE project (Guieu et al.,
2014a) showed that by stimulating predominantly heterotrophic bacteria, atmospheric dust
deposition can enhance the heterotrophic behavior of these oligotrophic waters. This has the
potential to reduce the fraction of organic carbon that can be exported to deep waters during the
winter mixing period (Pulido-Villena et al., 2008) and ultimately limit net atmospheric $CO_2$
drawdown.
Another effect induced by Saharan dust deposition is the export of particulate organic
carbon (POC) as lithogenic particles can aggregate and ballast dissolved organic matter (Bressac et





al., 2014; Desboeufs et al., 2014; Louis et al., 2017a; Ternon et al., 2010). This so-called lithogenic
carbon pump can represent a major part of the carbon export following a dust deposition event (up
to 50% during the DUNE experiment; Bressac et al., 2014). Recently, Louis et al. (2017a) showed
that Saharan dust deposition triggers the abiotic formation of transparent exopolymeric particles
(TEP), leading to the formation of organic-mineral aggregates, a formation process that is highly
dependent on the quality and quantity of TEP-precursors initially present in seawater.
In response to ocean warming and increased stratification, open ocean nutrient cycles are
being and will be perturbed in the next decades with a high confidence of having regionally variable
impacts on primary producers (IPCC, 2019). Overall, LNLC areas are expected to expand in the
future (Irwin and Oliver, 2009; Polovina et al., 2008) due to lower nutrient supply from sub-surface
waters (Behrenfeld et al., 2006), likely further increasing the role of atmospheric deposition as a
significant source of new nutrients to surface waters. The ongoing warming and acidification of the
global ocean (IPCC, 2019), both also evidenced in the Mediterranean Sea (e.g. Kapsenberg et al.,
2017; The Mermex group, 2011) raise the question on whether plankton communities will react
differently to dust deposition in a warmer and acidified environment. Although dependent on
resource availability, it is well known that remineralisation by bacteria is subject to positive
temperature control (López-Urrutia and Morán, 2007). Under severe nutrient limitation, there is no
evidence that warming will lead to an enhancement of primary productivity (Marañón et al., 2018),
further pushing the balance towards net heterotrophy in oligotrophic areas.
With respect to ocean acidification, an *in situ* mesocosm experiment conducted during the
summer stratified period in the Northwestern Mediterranean Sea showed that the plankton
community was rather insensitive to this perturbation under strong nutrient limitation (Maugendre
et al., 2017, and references therein). This is coherent with results from Maugendre et al. (2015),
based on a batch experiment, showing that, under nutrient-depleted conditions in late winter, ocean

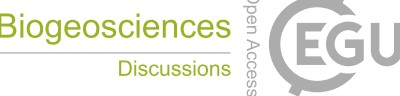

acidification has a very limited impact on the plankton community and that small species (e.g.
Cyanobacteria) might benefit from warming with a potential decrease of the export and energy
transfer to higher trophic levels. In contrast, in more eutrophic coastal conditions, Sala et al. (2016)
showed that ocean acidification exerted a positive effect on phytoplankton, especially on pico- and
nano-phytoplankton. Similarly, Neale et al. (2014) showed in a coastal ecosystem of the Alboran
Sea that ocean acidification could lead, although moderately, to high chlorophyll levels under low
light conditions with an opposite effect under high irradiance.
To date and to the best of our knowledge, there has been no attempts to evaluate the
behavior of plankton communities after the deposition of atmospheric particles in the context of
future levels of temperature and pH. Yet, following the recommendation from Maugendre et al.
(2017), any perturbation experiment for future climate conditions in the Mediterranean Sea should
consider atmospheric deposition as a source of new nutrients and consider both temperature and pH
as external forcings. Such experiments were conducted in the frame of the PEACETIME project
(ProcEss studies at the Air-sEa Interface after dust deposition in the MEditerranean sea;
http://peacetime-project.org/) during the cruise on board the R/V "Pourquoi Pas?" in May/June
2017. The project aimed at extensively studying and parameterizing the chain of processes
occurring in the Mediterranean Sea after atmospheric deposition, especially of Saharan dust, and to
put them in perspective of on-going environmental changes (Guieu et al., 2020). During that cruise,
three perturbation experiments were conducted in climate reactors (300 L tanks) filled with surface
water collected in the Tyrrhenian Sea (TYR), Ionian Sea (ION) and in the Algerian basin (FAST;
Fig. 1). Six tanks were used to follow simultaneously and with a high temporal resolution, the
evolution of biological activity and stocks, nutrients stocks, dissolved organic matter as well as
particles dynamics and export, following a dust deposition event simulated at their surface, both
under present environmental conditions and following a realistic climate change scenario for 2100



(ca. +3 °C and -0.3 pH units; IPCC, 2013). In this manuscript, we will present the general setup of
the experiments, the impacts of dust seeding and/or future climate change scenario on nutrients and
biological stocks. Among several other manuscripts related to these experiments that are introduced
here, a companion paper will be focusing on plankton metabolism (primary production,
heterotrophic prokaryote production) as well as on carbon budget.



## 2. Material and Methods

## 2.1. General setup

Six experimental tanks (300 L; Fig. 2) in which the irradiance spectrum and intensity can be finely controlled and in which future ocean acidification and warming conditions can be fully reproduced were installed in a temperature-controlled container. The tanks are made of high-density polyethylene (HDPE) and were trace-metal free in order to avoid contaminations, with a height of 1.09 m, a diameter of 0.68 m, a surface area of 0.36 m$^2$ and a volume of 0.28 m$^3$. All tanks were cleaned before the experimental work following the protocol described by Bressac and Guieu (2013). A weak turbulence was generated by a rotating PVC blade (9 rpm) in order to mimic natural conditions. Each tank was equipped with a lid containing six rows of LEDs (Alpheus©). Each of these rows were composed of blue, green, cyan and white units in order to mimic the natural sun spectrum. At the conical base of each tank, a polyethylene (PE) bottle collecting the exported material from above was screwed onto a polyvinyl chloride (PVC) valve that remained open during the duration of the whole experiment. Photosynthetically active radiation (PAR; 400-700 nm) and temperature were continuously monitored in each tank using respectively QSL-2100 Scalar PAR Irradiance Sensors (Biospherical Instruments©) and pt1000 temperature sensors (Metrohm©) connected to a D230 datalogger (Consort©).

The experimental protocol comprised two unmodified control tanks (C1 and C2), two tanks enriched with Saharan dust (D1 and D2) and two tanks enriched with Saharan dust and maintained under warmer (+3 ºC) and acidified (-0.3 pH unit) conditions (G1 and G2). The atmosphere above tanks C1, C2, D1 and D2 was flushed with ambient air (ca. 400 ppm, 6 L min$^{-1}$) and tanks G1 and G2 were flushed with air enriched with $CO_2$ (ca. 1000 ppm, 6 L min$^{-1}$) in order to prevent $CO_2$





degassing from the acidified tanks. $CO_2$ partial pressure ($pCO_2$) in both ambient air and
$CO_2$-enriched air was monitored using two gas analysers (LI-820, LICOR©). The $CO_2$
concentration in the $CO_2$-enriched air was manually controlled through small injections of pure $CO_2$
(Air Liquide©) using a mass flow controller.
Three experiments were performed at the long duration stations TYR, ION and FAST. The
tanks were filled by means of a large peristaltic pump (Verder© VF40 with EPDM hose, flow of
1200 L h$^{-1}$) collecting seawater below the base of the boat (depth of ~ 5 m), used to supply
continuously surface seawater to a series of instruments during the entire campaign. In order to
homogeneously fill the tanks, the flow was divided into six HDPE pipes distributing the water
simultaneously into the different tanks. Overall, the filling of the six tanks took ~2 h (including
rinsing and initial sampling, see thereafter). At the three stations, tanks were always filled at the end
of the day before the start of the experiments: TYR (17/05/2017), ION (25/05/2017) and FAST
(02/06/2017). While filling the tanks, this surface seawater was sampled for the measurements of
selected parameters (sampling time = t-12h, see Table 1). After filling the tanks, seawater was
slowly warmed using 500 W heaters, controlled by temperature-regulation units (COREMA©), in
G1 and G2 overnight to reach an offset of +3 °C. $^{13}$C-bicarbonate was added to all tanks at 4:00 am
(local time; Gazeau et al., in preparation, this issue) and G1 and G2 were acidified by addition of
$CO_2$-saturated filtered (0.2 μm) seawater (~1.5 L in 300 L; collected when filling the tanks at each
station) at 4:30 am to reach a pH offset of -0.3. Sampling for many parameters took place prior to
dust seeding (sampling time = t0, see Table 1). Dust seeding was performed between 7:00 and 9:00
(local time) in tanks D1, D2, G1 and G2. The same dust analog was used and the same dust flux
was simulated as for the DUNE 2009 experiments described in Desboeufs et al. (2014). Briefly, the
fine fraction (< 20 μm) of Saharan soils collected in southern Tunisia, which is a major source of
dust deposition over the northwestern Mediterranean basin, was used in the seeding experiments.

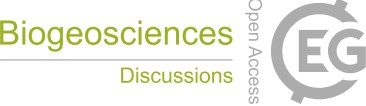

The particle size distribution showed that 99% of particles had a size smaller than 0.1 μm, and that
particles were mostly made of quartz (40%), calcite (30%) and clay (25%; Desboeufs et al., 2014).
This collected dust underwent an artificial chemical aging process by addition of nitric and sulfuric
acid ($HNO_3$ and $H_2SO_4$, respectively) to mimic cloud processes during atmospheric transport of
aerosol with anthropogenic acid gases (Guieu et al., 2010a, and references therein). To mimic a
realistic wet flux event of 10 g m$^{-2}$, 3.6 g of this analog dust were quickly diluted into 2 L of
ultrahigh-purity water (UHP water; 18.2 MΩ cm$^{-1}$ resistivity), and sprayed at the surface of the
tanks using an all-plastic garden sprayer (duration = 30 min). The N and P total contents in the dust
were 1.36 ± 0.09% of N and 0.055 ± 0.003% of P (see Desboeufs et al., 2014, for a full description
of dust chemical composition). The experimental protocol included the analysis of an extensive
number of biogeochemical parameters and processes, not all shown and discussed in this paper, that
are listed in Table 1. The experiment at stations TYR and ION lasted 72 h (3 days) whereas the last
experiment at station FAST was extended to four days. Seawater sampling was conducted 1 h (t1h),
6 h (t6h), 12 h (t12h), 24 h (t24h), 48 h (t48h) and 72 h (t72h) (+ 96 h = t96h for station FAST) after
dust addition. Acid-washed silicone tubes were used for transferring the water collected from the
tanks to the different vials or containers. For some parameters (e.g. nutrients, dissolved organic
carbon), sampled seawater was filtered online at the exit of the tanks on sterile membrane filter
capsules (gravity filtration with Sartobran© 300; 0.2 μm).

## 212 2.2. Analytical methods

## 213 2.2.1. Carbonate chemistry

Seawater samples for pH measurements were stored in 300 mL glass bottles with a glass
stopper, pending analysis on board (within 2 h). Samples were transferred to 30 mL quartz cells and





absorbances at 434, 578 and 730 nm were measured at 25 °C on an Cary60 UV-Spectrophotometer
(Agilent©) before and after addition of 50 µL of purified meta-cresol purple provided by Robert H.
Byrne (University of South Florida, USA) following the method described by Dickson et al. (2007).
pH on the total scale ($pH_T$) was computed using the formula and constants of Liu et al. (2011). The
accuracy of pH measurements was estimated to 0.007 pH units, using a TRIS buffer solution
(salinity 35, provided by Andrew Dickson, Scripps university, USA).

Seawater samples for total alkalinity ($A_T$; 500 mL) measurements were filtered on GF/F

membranes and analyzed onboard within one day. $A_T$ was determined potentiometrically using a
Metrohm© titrator (Titrando 888) and a glass electrode (Metrohm©, ecotrode plus) calibrated using
first NBS buffers (pH 4.0 and pH 7.0, to check that the slope was Nernstian) and then using a TRIS
buffer solution (salinity 35, provided by Andrew Dickson, Scripps university, USA). Triplicate
titrations were performed on 50 mL sub-samples at 25 ºC and $A_T$ was calculated as described by
Dickson et al. (2007). Titrations of standard seawater provided by Andrew Dickson (Scripps
university, USA; batch 151) yielded $A_T$ values within 5 µmol kg$^{-1}$ of the nominal value (standard
deviation = 1.5 µmol kg$^{-1}$, n = 40).

All parameters of the carbonate chemistry were determined from $pH_T$, $A_T$, temperature,

salinity, as well as phosphate and silicate concentrations using the R package seacarb[1]. Propagation
of errors on computed parameters was performed using the new function "error" of this package,
considering errors associated with the estimation of $A_T$, $pH_T$ as well as errors on dissociation
constants (Orr et al., 2018).
[1]  Seacarb: seawater carbonate chemistry with R. Gattuso, J.-P., J. M. Epitalon, H. Lavigne, J. C. Orr, B. Gentili, M.
Hagens, A. Hofmann, A. Proye, K. Soetaert and J. Rae, 2018. https://cran.r-project.org/package=seacarb



## 2.2.2. Nutrients

Seawater samples for dissolved nutrients were filtered online (<0.2 μm), collected in polyethylene bottles and immediately analyzed on board. Nitrate + nitrite ($NO_x$) and silicate ($Si(OH)_4$) measurements were conducted using a segmented flow analyzer (AAIII HR Seal Analytical©) according to Aminot and Kérouel (2007)with a limit of quantification of 0.05 μmol $L^{-1}$ for $NO_x$ and 0.08 μmol $L^{-1}$ for $Si(OH)_4$. In addition, for t-12h samples, the analysis of $NO_x$ was also performed by a spectrometric method in the visible at 540 nm, with a 1 m Liquid Waveguide Capillary Cell (LWCC). The limit of detection was ~10 nmol $L^{-1}$ and the reproducibility was ~6%. Also from samples taken at t-12h, the measurement of ammonium concentrations was performed on board using a Fluorimeter TD-700 (Turner Designs©) according to Holmes et al. (1999). This fluorimetric method is based on the reaction of ammonia with orthophtaldialdehyde and sulfite and has a limit of quantification of 0.01 μmol $L^{-1}$. Dissolved inorganic phosphorus (DIP) concentrations were quantified using the Liquid Waveguide Capillary Cell (LWCC) method according to Pulido-Villena et al. (2010). The LWCC was 2.5 m long and the limit of detection was 1 nmol $L^{-1}$.

## 2.2.3. Pigments

A volume of 2.5 L of sampled seawater was filtered onto GF/F filters, immediately frozen in liquid nitrogen and stored at -80 ºC pending analysis at the SAPIGH analytical platform at the Institut de la Mer de Villefranche (IMEV, France). Filters were extracted at -20 ºC in 3 mL methanol (100%) containing an internal standard (vitamin E acetate, Sigma©), disrupted by sonication and clarified one hour later by vacuum filtration through GF/F filters. The extracts were rapidly analyzed (within 24 h) on a complete Agilent© Technologies 1200 series HPLC system. The pigments were separated and quantified as described in Ras et al. (2008).


## 2.2.4. Flow cytometry

For the enumeration of autotrophic prokaryotic and eukaryotic cells, heterotrophic

prokaryotes and heterotrophic nanoflagellates (HNF) by flow cytometry, subsamples (4.5 mL) were

fixed with glutaraldehyde grade I 25% (1% final concentration), and incubated for 30 min at 4 °C,

then quick-frozen in liquid nitrogen and stored at -80 °C until analysis. Samples were thawed at

room temperature. Counts were performed on a FACSCanto II flow cytometer (Becton

Dickinson©) equipped with 3 air-cooled lasers: blue (argon 488 nm), red (633 nm) and violet (407

nm). The separation of different autotrophic populations was based on their scattering and

fluorescence signals according to Marie et al. (2010). *Synechococcus* spp. was discriminated by its

strong orange fluorescence (585 ± 21 nm), and pico- and nano-eukaryotes were discriminated by

their scatter signals of red fluorescence (> 670 nm). For the enumeration of heterotrophic

prokaryotes, cells were stained with SYBR Green I (Invitrogen – Molecular Probes) at 0.025% (vol

/ vol) final concentration for 15 min at room temperature in the dark. Stained prokaryotic cells were

discriminated and enumerated according to their right-angle light scatter (SSC) and green

fluorescence at 530/30 nm. In a plot of green versus red fluorescence, heterotrophic prokaryotes

were distinguished from autotrophic prokaryotes. For the enumeration of HNF, staining was

performed with SYBR Green I (Invitrogen—Molecular Probes) at 0.05% (v/v) final concentration

for 15-30 min at room temperature in the dark (Christaki et al., 2011). Cells were discriminated and

enumerated according to their SSC and green fluorescence at 530/30 nm. Fluorescent beads (1.002

μm; Polysciences Europe©) were systematically added to each analyzed sample as internal

standard. The cell abundance was determined from the flow rate, which was calculated with

TruCount beads (BD biosciences©). Biomasses of each group were estimated based on conversion

equations and/or factors found in the literature (see section 2.3).


## 2.2.5. Micro-phytoplankton and -heterotrophs

At t-12h (i.e. seawater sampled during the filling of the tanks), a volume of 500 mL was

sampled in glass vials and immediately preserved in a 5% acidic Lugol's solution pending analysis.

At the Laboratoire d'Océanographie de Villefranche (LOV, France), 100 mL aliquots were

transferred to sedimentation chambers (Utermohl) and counted under an inverted microscope at 200

to 400 magnifications.

## 2.2.6. Mesozooplankton

At the end of each experiment (t+72h for TYR and ION and t+96 h for FAST, after artificial

dust seeding), the sediment traps were removed, closed and stored with formaldehyde 4% (see

Gazeau et al., in preparation, this issue). The valve at the base of the tanks was then reopened to let

the remaining water inside the tanks (TYR 165-180 L; ION = 172.5 L and FAST = 150 L) pass

through a large PVC sieve (100 µm). The organisms retained on that mesh were gently removed

from the sieve using a washing bottle filled with filtered seawater (0.2 µm), and transferred directly

inside a 250 mL bottle. The bottle was filled with the sample (1/3 of the volume), and was

completed with formaldehyde 4% . The zooplankton digital images were obtained using a

ZooSCAN (Hydroptic©; Gorsky et al., 2010) at the PIQv-platform of EMBRC-France. The

identification of species was performed by automatic comparison with the library data set EcoTaxa

(https://ecotaxa.obs-vlfr.fr/, last access: 17/04/2020) and then all validated and corrected by a

human operator.



## 2.3 Computations

The maximum percentage of dissolution from dust observed with respect to N and P was calculated considering that these evapo-condensated dust contain $1.36 \pm 0.09\%$ of N and $0.055 \pm 0.003\%$ of P (Desboeufs et al., 2014). Based on maximal concentrations observed in the D and G tanks after seeding (two discrete sampling within 6 h), one can estimate the maximal % of dissolution of dust in seawater during the three experiments:

$$\%_{dissolution} = \frac{CONC_{max} - CONC_{init}}{CONC_{dust}} \cdot 100 \qquad (1)$$

where $CONC_{init}$ is the concentration of the corresponding nutrient in each tank before seeding (t0), $CONC_{max}$ corresponds to the concentration of the corresponding nutrient in each tank when nutrient concentration was at a maximum over the first 6 h after seeding as observed based on our discrete sampling procedure, and $CONC_{dust}$ corresponds to the maximum input of each nutrient, if 100% of its total concentration in the dust analog dissolves (as estimated based on dust chemical composition; Desboeufs et al., 2014; see above).

As micro-phytoplankton counting was not performed throughout the experiment, as a first approximation, autotrophic biomass was calculated as the sum of carbon contained in *Synechococcus*, pico-eukaryotes and nano-eukaryotes (abundances based on flow cytometry) and is therefore restricted to the fraction < 20 μm. For *Synechococcus*, conversion to carbon units were done considering 250 fg C cell$^{-1}$ (Kana and Glibert, 1987), while the equation proposed by Verity et al. (1992; $0.433 \, BV^{0.863}$ where BV refers to the biovolume) was used for pico- and nano-eukaryotes assuming a spherical shape and a diameter of 2 and 6 μm for the two groups, respectively. Percentages of these different groups were calculated in order to estimate the composition of the communities at the start and its evolution during the experiments. Furthermore, heterotrophic





biomass was computed as the sum of heterotrophic prokaryotes biomass and heterotrophic
nanoflagellates biomass. For heterotrophic prokaryotes, conversion to carbon units were done
considering 20 fg C cell$^{-1}$ (Lee and Fuhrman, 1987) and for heterotrophic nanoflagellates assuming
220 fg C $\mu m^{-3}$ (Børsheim and Bratbak, 1987), a spherical shape and a diameter of 3 μm. The ratio of
autotrophic and heterotrophic biomass during the experiments was used to evaluate the trophic
status of the investigated communities and its evolution. Finally, a proxy for micro-phytoplankton
biomass ($B_{micro}$) was estimated following Vidussi et al. (2001), as the sum of Fucoxanthin and
Peridinin.



# 3. Results

## 3.1. Initial conditions

Initial conditions of various measured parameters at the three sampling stations while filling the tanks are shown in Table 2. $pH_T$ and total alkalinity concentrations observed when pumping seawater for the experiments (before $^{13}$C-bicarbonate addition and dust seeding: t-12h) followed a west to east increasing gradient (8.03, 8.04 and 8.07; 2443, 2529 and 2627 µmol kg$^{-1}$ at FAST, TYR and ION, respectively). $NO_x$ concentrations were maximal at station FAST with a $NO_x$:DIP molar ratio of ~ 4.6. Very low $NO_x$ concentrations were observed at stations TYR and ION (14 and 18 nmol L$^{-1}$, respectively). DIP concentrations were the highest at station TYR (17 nmol L$^{-1}$) and the lowest at the most eastern station (ION, 7 nmol L$^{-1}$). Consequently, the lowest $NO_x$:DIP ratio was measured at TYR (0.8), compared to ION and FAST (2.8 and 4.6, respectively). Ammonium concentrations were maximal at TYR (0.045 µmol L$^{-1}$), intermediate at ION (0.022 µmol L$^{-1}$), and minimal at FAST (below detection limit). Silicate concentrations were similar at stations TYR and ION (~ 1 µmol L$^{-1}$) and higher than at station FAST (0.64 µmol L$^{-1}$).

Very low and similar concentrations of chlorophyll *a* were measured at the three stations (0.063 - 0.072 µg L$^{-1}$). The proportion of the different major pigments (Fig. 3) showed that phytoplankton communities at stations TYR and ION were very similar with a dominance of Prymnesiophytes (i.e. 19'-hexanoyloxyfucoxanthin; Ras et al., 2008) followed by Cyanobacteria (i.e. Zeaxanthin; Ras et al., 2008). In contrast, at station FAST, the planktonic community was clearly dominated by photosynthetic prokaryotes (i.e. Zeaxanthin and Divinyl-chlorophyll *a*; Cyanobacteria and Prochlorophytes, respectively; Ras et al., 2008). At all three stations, the

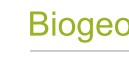
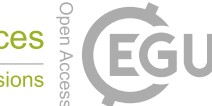

proportion of pigments representative of larger species (i.e. Fucoxanthin and Peridinin; diatoms and
dinoflagellates respectively; Ras et al., 2008) were very small (< 5%).
Cellular abundances of all studied microorganisms (phytoplankton, micro-grazers,
heterotrophic bacteria) were the highest at FAST (Table 2). Picoeukaryotes, *Synechococcus* and
heterotrophic prokaryotes abundances followed an east to west increasing trend (ION < TYR <
FAST). In contrast, nano-eukaryotes abundance was similar at FAST and ION, and minimal at
TYR. The abundance of heterotrophic nanoflagellates (HNF) were similar at TYR and FAST
(~110-125 cells mL$^{-1}$), twice as high as the one observed at station ION. This east to west increasing
trend was also observed for micro-phytoplankton and micro-heterotrophs abundances (microscopic
analyses; Table 2). The ratio between autotrophic biomass and heterotrophic biomass was clearly in
favor of the heterotrophic compartment at stations TYR and FAST (~0.6 at the two stations) but the
opposite was found at station ION (ca. 1.3).
## 3.2. Experimental conditions
Irradiance levels during the experiments in controls were maximal at station ION and
minimal at station FAST (daily average maximum levels in controls: ~ 1050, ~ 1130 and ~ 1020
µmol photons m$^{-2}$ s$^{-1}$ at TYR, ION and FAST, respectively; Fig. 4). Decreases of water transparency
after dust addition was observed at all three stations with a maximum dust impact at station ION
and the lowest impact at station FAST where irradiance levels decreased by only 60 µmol photons
m$^{-2}$ s$^{-1}$ after dust addition (average between tanks D and G). At station TYR, a more pronounced
decrease was observed in acidified and warmed tanks (G1 and G2) with a decrease of daily average
maximum irradiance of ~ 60 and ~ 160 µmol photons m$^{-2}$ s$^{-1}$ as compared to dust-amended tanks D
and controls, respectively. Temperature control (Fig. 4) was not optimal showing deviations
between replicates of treatment G of up to 1.5 °C (station ION). Temperature in controls and D



tanks displayed a daily cycle with an increase during the day and a decrease at night. Overall, the
differences between the warmed treatment (G) and the other tanks were +3, +3.2 and +3.6 °C at
TYR, ION and FAST, respectively.
Addition of $CO_2$-saturated filtered seawater led to a decrease of $pH_T$ from 8.05 ± 0.004
(average ± SD between C1, C2, D1 and D2 at t0) to 7.74 (average between G1 and G2) at station
TYR, from 8.07 ± 0.002 to 7.78 at station ION and 8.05 ± 0.001 to 7.72 at station FAST (Fig. 5).
$pH_T$ levels remained more or less constant in ambient pH levels tanks during all three experiments
with no clear impact of dust addition in tanks D1 and D2. In lowered pH tanks, pH levels gradually
increased during the experiments with a systematic larger increase in one of the duplicates (G1). Yet
$pH_T$ increases remained moderate thanks to the flushing of $CO_2$-enriched air above the tanks ($pCO_2$
of 1017 ± 11, 983 ± 96, 1023 ± 25 ppm at TYR, ION and FAST, respectively; data not shown).
Partial pressure of $CO_2$ in ambient air was similar at the three stations, i.e. 410 ppm (data not
shown). At all three stations, $^{13}C$-addition led to an increase of total alkalinity between 6 and 11
µmol $kg^{-1}$ and dust addition led to a decrease in tanks D and G between 8 and 16 µmol $kg^{-1}$ with no
apparent effects of warming and acidification. Overall, no large changes in this parameter were
observed during the experiments (Fig. 5).

## 392 3.3. Changes in nutrient concentrations

Dust addition in tanks D and G led to a rapid and maximum input of $NO_x$ (as observed
during the first 6 h; Fig. 6; Table 3) of ~ 11 µmol $L^{-1}$ at all three stations with no differences
between both treatments. The corresponding dissolution percentage of N contained in the dust
analog was between 94 and 99%. In contrast, maximum DIP release (within 6 h after dust addition)
from the dust analog was much smaller and comprised between 20 and 37 nmol $L^{-1}$, with slightly
higher release at FAST (31-37 nmol $L^{-1}$) as compared to the other stations. Dissolution percentages





for DIP were estimated between 9.2 and 17.3% of total phosphorus contained in dust. As a
consequence of these contrasted dissolution of N and P, $NO_x$:DIP ratios increased from initial
values below 5 to above 300, within 6 h after dust seeding, in the dust amended (D and G) tanks
(Fig. 6).
After these rapid increases due to N and P releases in dust amended tanks, both variables
decreased with time. While nutrient variability was small in control tanks over the duration of the
experiments ($NO_x$ and DIP variations below 20 and 3 nmol $L^{-1}$, respectively), large decrease of both
elements was measured in dust amended tanks (D and G; Table 4). For $NO_x$, similar linear
decreases were observed throughout the experiments at stations TYR and ION with no visible
differences between tanks D and G. In contrast, at station FAST, a more pronounced decrease in
$NO_x$ was observed in dust-amended (D and G) tanks as compared to the other stations, with
detectable larger decreases in warmed and acidified tanks relative to the D treatment. Nevertheless,
at all stations, $NO_x$ concentrations in D and G treatments remained far above ambient levels
throughout the experiments ($> 9$ μmol $L^{-1}$). Abrupt decreases in DIP were observed during the three
experiments after the initial increase. At station TYR, after 24 h, all DIP released from dust
decreased to initial levels in tanks G while it took two more days to reach initial levels in tanks D.
In contrast, at station ION, no clear difference in DIP dynamics was observed between treatments D
and G, with concentrations that decreased rapidly during the first 24 h but that remained above
initial levels until the end of the experiment. At station FAST, similarly to station TYR, DIP
decreased rapidly from t12h in treatment G, reaching levels close to initial conditions at the end of
the experiment. DIP decrease was much lower in treatment D (Table 4) with concentrations
maintained far above ambient levels throughout the experiment. As a consequence of these
differences between $NO_x$ and DIP dynamics as well as differences among stations, $NO_x$:DIP ratio





increased during the experiments with clear differences between stations (Fig. 6) and remained
much higher than that in the controls over the duration of the three experiments.

Silicate dynamics showed at all stations higher concentrations in dust amended (D and G)

tanks relative to the controls. At TYR, while in control tanks, concentrations remained stable, they
increased linearly with time in the other tanks (D and G) with no apparent effect of the imposed
increase in temperature and decrease in pH (i.e. tanks G). Difference of $Si(OH)_4$ concentration
between dust amended treatments (D and G) and controls was ~0.1 $\mu mol\ L^{-1}$ at the end of the
experiment. At station ION, after an initial decrease of concentrations between t-12h and t0,
concentrations increased in all tanks until the end of the experiment with higher concentration in
dust amended tanks (D and G) than in controls (no difference between D and G treatments). In
contrast, at FAST, concentrations increased between t-12h and t0, and continued to increase in all
tanks (with higher values in dust amended tanks) until t48h and then decreased until the end of the
experiment. At the end of the experiment (t96h), $Si(OH)_4$ concentration was higher in the G
treatment than in the D treatment which was similar to the controls.

## 3.4. Changes in biological stocks


Regarding biological stocks, temporal dynamics showed very different patterns with respect

to the sampling station. At TYR, total chlorophyll *a* concentrations did not change in dust amended
tanks maintained under ambient levels of temperature and pH (Fig. 7) and even led to slightly
decreased values 24 h after dust addition (e.g. -35 to -38% in D1 and D2, respectively as compared
to controls; Table 5). No clear effect of dust addition (tanks D vs. C) were detectable for all groups
based on pigment analyses (Fig. 7). Results obtained based on flow cytometry counting (Fig. 8)
were coherent with these observations and showed stronger decreases in cell abundances for < 20
µm autotrophic groups in tanks D1 and D2 (-77 to -80%). In contrast, at this station, the abundance



of heterotrophic prokaryotes (HP) increased rapidly after dust addition both under ambient
(+53-68%) and future (+68%) environmental conditions, with no clear difference among those
treatments. In warmed and acidified tanks, strong discrepancies between the duplicates were
observed for pigments and autotrophic cell abundances. Indeed, tank G1 showed moderately higher
levels for all variables as compared to tanks C at the exception of pico-eukaryotes, while in G2 all
variables responded strongly to dust addition with maximum relative changes of > 300% (at the
exception of nano-eukaryotes: +119%). While HNF abundances responded positively to the
treatments in D1, D2 and G2 (+100-352%), abundances increased sharply in tank G1 towards the
end of the experiment (+1095%). At ION, a clear distinction between treatments could be observed
for almost all pigments and cell abundances (Fig. 7, Fig. 8). At the exception of nano-eukaryotes
and HNF, all variables (pigments and cell abundances) increased as a response to both dust addition
and warmed/acidified conditions (i.e. C < D < G). As an example (Table 5), the maximum relative
changes as compared to controls observed for total chlorophyll *a* were 109-183% and 399-426% in
tanks D and G, respectively. The highest stimulation to dust addition was observed for
*Synechococcus* with a +317-390% increase and +805-1425% increase in D and G tanks respectively
(Table 5). Abundances of nano-eukaryotes and HNF suggested no impact of dust addition under
ambient conditions but a positive impact in treatment G. In contrast to what was observed at TYR
for HP abundances, an effect of temperature and pH was observed at station ION with a higher
impact of dust addition under future environmental conditions. At station FAST, all above
mentioned variables related to biological stocks increased strongly after dust addition (Fig. 7, Fig. 8
and Table 5). For instance, total chlorophyll *a* increased following an exponential trend until the end
of the experiment reaching maximal values at t96h with slightly lower values observed under
ambient environmental conditions (+237-318% in D tanks vs. ~ +400% in G tanks).
Prymnesiophytes (i.e. 19'-hexanoyloxyfucoxanthin) and diatoms (i.e. Fucoxanthin) appeared as the





groups benefiting the most from dust addition with no large impacts of warming/acidification. In
contrast, Pelagophytes (i.e. 19'-butanoyloxyfucoxanthin) and green algae (i.e. Total Chlorophyll *b*)
responded much more in treatment G than in treatment D. Finally, although Cyanobacteria (i.e.
Zeaxanthin) responded faster to dust addition under future environmental conditions (tanks G), this
effect tended to attenuate towards the end of the experiment. In contrast to estimates based on
HPLC data, increases in cell abundances did not generally take place until the end of the
experiment. While abundances in pico-eukaryotes increased until t96h in treatment D, abundances
sharply declined between t72h and t96h for this group in treatment G. The same trend was observed
for *Synechococcus* during this experiment, although discrepancies between duplicates in treatment
D at sampling time t96h did not allow drawing conclusions on the behavior of this group at the end
of the experiment. Both under ambient and future conditions, abundances of nano-eukaryotes
declined sharply between t72h and t96h. The decline in HP abundances appeared even earlier
during the experiment with moderate maximum relative differences as compared to controls
observed at t48h. HP abundances declined very sharply between t48h and t96h in treatment G,
reaching control levels, while this decline was less sharp under ambient environmental levels.
Finally, HNF dynamics during this experiment was hard to evaluate with no clear effects of dust
addition or pH/temperature conditions and with a large increase in abundances in only one duplicate
of treatment G (t24h) followed by a gradual decrease.
Abundances of meso-zooplankton at the end of the experiments showed relatively similar
values at stations TYR and ION while much higher levels were observed at station FAST (Fig. 9).
As a consequence of large variability between duplicates at stations TYR and ION, no clear effects
of treatments were detected. At station FAST, although the sample size was too low to statistically
test for differences, higher total abundances of meso-zooplankton species were observed in the
dust-amended tanks with no differences between ambient and future conditions of temperature and



pH. However, differences in abundance were visible between these two treatments for specific
groups, with respectively higher abundance of Harosa and lower abundance of Crustacea (other
than copepods) and Mollusca in warmed and acidified tanks.



# 4. Discussion

## 4.1. Initial conditions

Overall, the three experiments were conducted with surface seawater collected during typical stratified oligotrophic conditions typical of the open Mediterranean Sea at this period of the year. However, at all three stations, initial concentrations of $NO_x$ (14, 18 and 59 nmol $L^{-1}$ at TYR, ION and FAST, respectively; Table 2) were lower that the ones reported by Manca et al. (2004) in surface waters (5 m) in these areas in spring (0.036 ± 0.10, 0.275 ± 0.358 and 0.183 ± 0.282 μmol $L^{-1}$ for the areas corresponding to TYR, ION and FAST, respectively; http://doga.ogs.trieste.it/medar/climatologies/, last access: 28/04/2020). Similarly, surface DIP concentrations as measured at the three stations were lower than values extracted from the compilation of Manca et al. (2004) for the same period (0.072 ± 0.072, 0.054 ± 0.035 and 0.115 ± 0.078 μmol $L^{-1}$ in the areas corresponding to TYR, ION and FAST, respectively). However, direct measurements of $NO_x$ and DIP concentrations using nanomolar techniques (as performed in our study) are scarce in the Mediterranean Sea, limiting our ability to compare our results with these published values which, in any case, show large interannual variability. Djaoudi et al. (2018) reported low DIP values in the three studied basins. Furthermore, low observed concentrations of $NO_x$ and DIP at all three stations during our study were also in agreement with reported concentrations in the coastal waters of Corsica during experiments using *in situ* mesocosms in June, whether during the DUNE project (DIP ~5 nmol $L^{-1}$; Pulido-Villena et al., 2014; $NO_x$ < 30 nmol $L^{-1}$; Ridame et al., 2014) or during the MedSeA project ($NO_x$ ~ 50 nmol $L^{-1}$ and DIP ~ 35 nmol $L^{-1}$; Louis et al., 2017b). Furthermore, at all three stations, $NO_x$:DIP molar ratios were well below the Redfield ratio (16:1) and are consistent with ratios found in these previously cited studies. Both low $NO_x$:DIP ratio and low nutrient concentrations suggest that communities found at the three stations



experienced N and P co-limitation at the start of the experiments, as previously shown by Tanaka et
al. (2011). Some enrichment experiments in DIP, $NO_3$+$NH_4$, glucose, alone or in combinations were
conducted using seawater sampled while filling the tanks. Bacterial production was mainly
stimulated by N+P addition at the three sites, although a slight stimulation was also detected after P
addition alone at TYR and ION (France Van Wambeke, pers. comm.). Initial concentrations of
dissolved Fe in the sampled seawater ranged from 1.5 nmol $L^{-1}$ (TYR) to 2.5 nmol $L^{-1}$ (ION;
Roy-Barman et al., in preparation, this issue). Such concentrations were unlikely limiting for
biological activity as previously shown in the Mediterranean Sea (Bonnet et al., 2005; Ridame et
al., 2014).

Total chlorophyll *a* concentrations of ~ 0.06 - 0.07 μg $L^{-1}$ (Table 2) were typical of

chlorophyll *a* levels found in these areas of the surface Mediterranean Sea at this period of the year,
as seen by satellite (Bosc et al., 2004), or from a database of *in situ* measurements (Manca et al.,
2004). During the DUNE and MedSeA projects cited above, chlorophyll *a* concentrations around
0.07 μg $L^{-1}$ were also encountered at the start of these experiments conducted in coastal waters
(Gazeau et al., 2017; Ridame et al., 2014). Although total chlorophyll *a* concentrations were rather
similar between the three tested stations, the composition of the phytoplankton communities, based
on HPLC pigment analyses, differed substantially. Indeed, while the communities were dominated
by nano-eukaryotic species at stations TYR and ION, both HPLC and flow cytometry data suggest
a larger contribution of pico-eukaryotes and Cyanobacteria at station FAST. Micro-autotrophs (e.g.
large diatoms and dinoflagellates) were slightly higher at station FAST. Due to their low
competitiveness during periods of nutrient limitation, the small contribution of large phytoplankton
cells at the start of the experiment is a fingerprint of LNLC areas and surface Mediterranean waters
at this period of the year (Siokou-Frangou et al., 2010). Autotrophic biomasses, as estimated based
on flow cytometry data (see Material and Methods) were similar at station TYR and ION (5.6 and



6.0 µg C L$^{-1}$) and maximal at FAST (7.7 µg C L$^{-1}$; Table 2). Although these estimates do not take
into account the contribution of micro-autotrophs, they appear to be in fair agreement with
estimates based on total chlorophyll *a* data, assuming a carbon to chlorophyll ratio of 70 (Bellacicco
et al., 2016), i.e. 4.4, 4.6 and 5.1 µg C L$^{-1}$ at TYR, ION and FAST, respectively. Furthermore, as
already mentioned, based on pigment analyses (HPLC), the sum of Fucoxanthin and Peridinin
(representative of diatoms and dinoflagellates, respectively) represented only ~10% of the total
chlorophyll *a* biomass at all stations. As biomass of both heterotrophic nanoflagellates and
prokaryotes followed a west to east gradient (FAST > TYR > ION), ratio of autotrophic vs
heterotrophic biomass appeared clearly in favor of the heterotrophic compartment at stations TYR
and FAST (ratio of 0.6) while a value above the metabolic balance was estimated at ION (ratio of
1.3). This is coherent with the highest net community production (NCP) rates being reported at this
station by Gazeau et al. (in preparation, this issue) showing that the initial community at the start of
this experiment was very close to metabolic balance (mean ± SE: -0.06 ± 0.09 µmol O$_2$ L$^{-1}$ d$^{-1}$). The
highest community respiration rates and consequently lowest NCP rates were measured at station
TYR (-1.9 µmol O$_2$ L$^{-1}$ d$^{-1}$) further suggesting that the autotrophic plankton community was not
very active (Ridame et al., in preparation, this issue) also confirmed by the lowest rate of CO$_2$
fixation; Ridame et al., in preparation, this issue), and relying on regenerated nutrients, as shown by
the highest level of NH$_4^+$ measured at the start of this experiment. In contrast, the community at
station FAST although slightly heterotrophic (Gazeau et al., in preparation, this issue) and limited
by the low amount of nutrients (Table 2) was the most active as shown by the highest levels of $^{14}$C
production and heterotrophic prokaryote production (Gazeau et al., in preparation, this issue) as
well as N$_2$ fixation (Ridame et al., in preparation, this issue). Altogether, the heterotrophic signature
of the three investigated stations, although closer to metabolic balance for ION, reflected typical
natural biogeochemical conditions in the Mediterranean Sea during late spring to early summer
(Regaudie-de-Gioux et al., 2009).

## 568 4.2. Experimental assessment

The experimental tanks used in this study have already been validated in several studies
designed to investigate the inputs of macro- and micro-nutrients (e.g. $NO_x$, DIP, DFe) and the
export of organic matter, under close-to-abiotic conditions (seawater filtration onto 0.2 μm)
following simulated wet dust events using the same analog as used in our study (Bressac and Guieu,
2013; Louis et al., 2017a, 2018). Louis et al. (2017a, 2018) further investigated these impacts under
lowered pH conditions. During these experiments, no control of atmospheric $p$CO$_2$ was performed
and pH levels in the acidified filtered seawater rapidly increased due to CO$_2$ degassing (from ~7.4
to ~7.7 in six days). Prior to the cruise, we improved our experimental system to allow mimicking
future conditions by controlling atmospheric $p$CO$_2$ in addition to light and temperature (i.e. climate
reactors). During our experiments, thanks to the control of atmospheric $p$CO$_2$ (~ 1000 ppm), we
significantly reduced CO$_2$ degassing and maintained pH levels close to experimental targets.
However, as can be seen in Fig. 5, the regulation was consistently more efficient in tank G2 as
compared to G1. We attribute this small discrepancy (highest difference of 0.04 pH units between
the two G tanks at FAST) to a potential leak or a longer flushing time above tank G1. Nevertheless,
we do not anticipate this as an issue.
The lids above tanks, equipped with LEDs in order to reproduce sunlight intensity and
spectrum, were used for the first time during these experiments. The maximal intensity reached
under control conditions (C1, C2) was between 900 and 1000 μmol photons m$^{-2}$ d$^{-1}$. Although
slightly lower than estimates for the Northwestern Mediterranean Sea at 5 m depth in June (~1100
μmol photons m$^{-2}$ d$^{-1}$; Bernard Gentili, personal communication, 2017), simulated intensities were



fairly consistent between duplicates under control conditions (C1, C2) and under dust-amended
conditions (D1, D2). In contrast, larger differences were observed between warmed and acidified
tanks (G1 and G2; maximal differences of 100-200 μmol photons m$^{-2}$ d$^{-1}$ depending on the
experiment) that generally increased during each experiment. The reasons of these discrepancies are
not clear and could result from differences of light intensity generated by the lids, of PAR sensors
sensitivity and/or of the amount of particles remaining in the tanks. Unfortunately, although
replication appeared satisfactory for this treatment (except at station TYR; see below), we can not
fully exclude a potential impact of these technical issues on our results for this warmer and acidified
treatment. A similar conclusion can be drawn regarding temperature regulation in the container
where temperature was not spatially homogeneous, leading to significant differences among
replicates. After this study, experimental tanks were installed in a new container in order to solve
these problems.
The experimental strategy chosen during this study implied considering three different
treatments: control, simulation of dust deposition and simulation of dust deposition under future
projected environmental conditions. This unbalanced design, i.e. without the consideration of a
treatment without dust addition under future temperature and pH levels, was chosen for practical
reasons as only six tanks could be used for this study. Furthermore, as already mentioned, previous
experiments clearly showed very limited effects of these drivers when communities are strongly
limited by nutrient availability (Maugendre et al., 2017), therefore the objective of our study was to
test the impact of an external forcing (atmospheric deposition) under future conditions, without
discriminating warming from acidification effects. More importantly, the relatively low number of
experimental units that could be installed in an ambarcable clean container, implied considering
duplicated tanks for each treatment. This forced choice implied the impossibility to perform
statistical analyses on our results, as at least triplicates are necessary for most statistical tests.



Differences between duplicates were, for the vast majority of studied variables and processes, lower
than differences between treatments and appear acceptable considering the difficulty to incubate
plankton communities for which slight differences in their initial composition can translate into
very important differences in dynamics (Eggers et al., 2014). Unfortunately, while no large
replication issues were observed during the ION and FAST experiments, very important
discrepancies were detected for tanks of the warmed and acidified treatment at station TYR. The
reasons behind the different behavior of the autotrophic community in tanks G1 and G2 are not
fully understood but we strongly suspect that heterotrophic nano-flagellates, feeding mainly on
prokaryotic picoplankton (Sherr and Sherr, 1994), exerted a strong top-down control on this group
in tank G1 in which HNF abundance sharply increased during the experiment (+1100% in G1 vs. +
300% in G2). Interestingly, while autotrophic prokaryotes were clearly impacted, no differences
between the two tanks G1 and G2 were observed for heterotrophic prokaryotes although
nanoflagellates are known to feed to this group as well (Sherr and Sherr, 1994). Heterotrophic
nano-flagellates were likely not the only group of grazers which abundance increased during this
experiment in tank G1 as the biomass of diatoms (i.e. Fucoxanthin) did not increase in this tank.
Nevertheless, as no analyses of micro-grazer abundances were performed during the experiments,
this hypothesis can not be verified. All in all, these discrepancies for this treatment at station TYR
remain an issue and prevent us from drawing any conclusion on the combined effect of temperature
and pH on the dynamics of the community for that station.

## 632 4.3. Impact of dust addition

During the three experiments, the observed increases in $NO_x$ and DIP few hours after dust
addition were rather similar to the enrichment levels obtained during the DUNE experiments at the
surface of the mesocosms (~ 50 m³) after the simulation of a wet dust deposition using the same



dust analog and the same simulated flux (Pulido-Villena et al., 2014; Ridame et al., 2014). $NO_x$
levels moderately decreased over the course of our experiments due to biological uptake (50-1420
nmol $L^{-1}$, depending on the experiment). The opposite feature was observed for the DIP released by
dust that rapidly decreased during our experiments except at station FAST in the D treatment where
final concentrations did not reach initial levels. These enrichment levels, especially for $NO_x$, were
much higher than those observed by Pitta et al. (2017, and references therein) during land-based
mesocosm experiments in the Eastern Mediterranean Sea, in which a dry Saharan deposition was
simulated. In contrast to this experiment, the objective of our study was to assess the impact of wet
dust deposition, the main dust deposition pathway in the Western Mediterranean Sea (Loÿe-Pilot
and Martin, 1996). Furthermore, following observations of mixing between dust and polluted air
masses during their transport (e.g. Falkovich et al., 2001; Putaud et al., 2004), we chose to use an
evapo-condensed dust analog that mimics the processes taking place in the atmosphere prior to
deposition, essentially the adsorption of inorganic and organic soluble species (e.g. sulfate and
nitrate; see Guieu et al., 2010a, for further details). The imposed evapo-condensation processes are
responsible for the large nitrate releasing capacity of the dust particles used in our study. Regarding
the intensity of simulated wet deposition event, the 10 g $m^{-2}$ deposition event considered here
represents a high but realistic scenario, as several studies reported even higher short deposition
events in this area of the Mediterranean Sea (Bonnet and Guieu, 2006; Loÿe-Pilot and Martin, 1996;
Ternon et al., 2010).

Although $NO_x$ and DIP increases after dust addition were rather similar during our three

experiments, interestingly the impacts on plankton community composition and functioning were
drastically different. Most experiments reporting on the effect of dust addition in the Mediterranean
Sea showed significant increases in chlorophyll *a* concentrations (mean 89% increase; Guieu and
Ridame, 2020). Such fertilization of primary producers was indeed observed at stations ION and

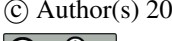



FAST under present conditions (maximum change in total chlorophyll *a* relative to the controls was
~280% at FAST and ~150% at ION). The largest increase in chlorophyll *a* concentrations at station
FAST is coherent with the largest observed $NO_x$ decrease following dust addition at this station.
Interestingly, following dust addition at this station, autotrophic production did not lead to DIP
exhaustion throughout the experiment as DIP concentrations were still above ambient conditions at
the end of the experiment. Maximal primary production rates ($^{14}$C-incorporation) at this station at
the end of the experiment suggest a strong DIP recycling and the dominance of regenerated
production towards the end of the experiment (Gazeau et al., in preparation, this issue). Guieu et al.
(2014b) showed, based on the analysis of eight aerosols addition studies, that *Synechococcus* had in
most of the cases a weak responses to aerosol addition in contrast to nano- and
micro-phytoplankton, suggesting that aerosol deposition may lead to an increase in larger size class
phytoplankton. Yet, *Synechococcus* were well stimulated in some dust addition experiments (Herut
et al., 2005; Lagaria et al., 2017; Paytan et al., 2009), similar to what was observed at both stations
ION and FAST, where *Synechococcus* abundance was clearly enhanced by dust deposition. The
increase in *Synechococcus* abundance to dust-amended tanks was the highest relative to those of
pico- and nano-eukaryotes at these stations. This was especially true at station ION where no clear
response to nutrient enrichment was observed for nano-eukaryotes throughout the experiment.
However, it must be stressed that our experiments were performed over a relatively short period (3
to 4 days), and the sharp increase in Fucoxanthin paralleled by a decrease in silicates, at the end of
the experiment at station FAST where DIP limitation was not yet apparent, suggests a delayed
response of diatoms as compared to smaller groups (i.e. autotrophic prokaryotes, pico- and
nano-eukaryotes). Although this was not observed based on pigment analyses, the sharp decline in
nano-eukaryote abundances at the end of the FAST experiment following dust addition, further

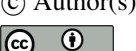



suggests that this group, reacting quickly to nutrient enrichment was progressively grazed and/or
outcompeted by larger phytoplankton species.

In contrast to what was observed at stations ION and FAST, no stimulation of autotrophic

biomass and primary production rates (Gazeau et al., in preparation, this issue) was observed in the
dust treatments under present conditions at station TYR. To the best of our knowledge, this is the
first experimental evidence of a complete absence of response from an autotrophic community
following dust wet deposition. There is clear evidence that not only phytoplankton but also
heterotrophic bacteria are limited by inorganic nutrients, mainly DIP, in oligotrophic systems
(Obernosterer et al., 2003; Wambeke et al., 2002), thus suggesting that the supply of these resources
could explain variability in bacterial activity. Many recent studies have shown significant increase
in heterotrophic bacterial abundance, respiration and/or production following dust deposition in
oligotrophic ecosystems (Lekunberri et al., 2010; Pitta et al., 2017; Pulido-Villena et al., 2008,
2014; Romero et al., 2011). Most of the time, heterotrophic processes appear to be more stimulated
by dust pulses compared to autotrophic processes with increasing degree of oligotrophy, the
dominant response being modulated by the competition for nutrients between phytoplankton and
bacteria (Marañón et al., 2010). This is clearly what was observed at this station, with heterotrophic
prokaryotes reacting quickly and strongly to nutrient addition both in terms of abundances (max: +
53-68%) and production rates (max: + 787-946%; Gazeau et al., in preparation, this issue). The
absence of response from autotrophic stocks could be due to a tight top-down control from grazers
hiding potential responses from the autotrophic community (Lekunberri et al., 2010; Marañón et al.,
2010). Feliu et al. (2020, this issue) have shown that the mesozooplankton assemblage at TYR was
clearly impacted by a dust event that took place nine days before sampling at that station (François
Dulac, Pers. Com. 2019) and evidenced by dust export in *in situ* deployed sediment traps (Bressac
et al., in preparation, this issue). This dust deposition likely stimulated phytoplankton growth and



consequently increased the abundance of herbivorous grazers (copepods) and attracted carnivorous
species. After the rapid increase observed a few hours after dust addition, DIP levels decreased to
reach similar levels than in control tanks at the end of this experiment (Fig. 6). Yet, heterotrophic
prokaryote abundances increased until the end of the experiment (Fig. 8) although production rates
reached a plateau after 24 h (Gazeau et al., in preparation, this issue). This is coherent with
measurements of the alkaline phosphatase activity that slightly increased at the end of the
experiment in dust-amended tanks suggesting the use of dissolved organic phosphorus by bacteria
to compensate for the increasing lack of DIP (Gazeau et al., in preparation, this issue). Altogether,
the strong stimulation of heterotrophic prokaryotes and the absence of detectable effects on the
autotrophic compartment drove the community towards a stronger net heterotrophic state as shown
by increases in community respiration and decreases in net community production rates in
dust-amended as compared to control tanks (Gazeau et al., in preparation, this issue).
At station FAST, the competition for nutrients between autotrophs and heterotrophs was
clearly in favor of autotrophs. While, as discussed above, all groups of primary producers benefited
from nutrient enrichment at this station, the increases in heterotrophic prokaryote abundances were
rather limited following dust deposition, leading to an increase of net community production rates
throughout this experiment to reach positive levels while control tanks remained below metabolic
balance (Gazeau et al., in preparation, this issue). At station ION, the situation was somewhat
intermediate with a parallel enhancement of both autotrophic and heterotrophic stocks and
processes, although the system was slightly in favor of net autotrophy at the end of the experiment
(Gazeau et al., in preparation, this issue).
Transfer of newly produced organic matter to higher trophic levels in the different
treatments was evaluated through the quantification of meso-zooplankton abundance at the end of
each experiment. Although we are fully aware that such an approach is certainly criticizable





considering the low incubation times (3 to 4 days), it may still be representative of lowered
mortality or faster growth. Altogether it does not appear as a surprise that an increase in
meso-zooplankton abundances was only detected at station FAST where the strongest enhancement
of primary production was observed. Such an increase in meso-zooplankton abundance in the
dust-amended as compared to control treatment was observed during land-based mesocosm
experiments in the Eastern Mediterranean Sea (Pitta et al., 2017).

Finally, although no clear effects of dust deposition under present conditions were detectable

on autotrophic prokaryotes at station TYR, the strongest increase in $N_2$ fixation rates was recorded
at this station (+434-503%, as compared to +173-256% and +41-49% at ION and FAST,
respectively; see Ridame et al., in preparation, this issue, for more details). However, the potential
impact of this process on $NO_x$ concentration is highly negligible compared to the very large stock of
$NO_x$ present in the dust-amended tanks, as less than 1 nmol $L^{-1}$ $d^{-1}$ of $NO_x$ can be produced by this
process (Ridame et al., in preparation, this issue).

## 744 4.4. Impact of warming and acidification

Very few past studies have investigated the release and fate of nutrients from atmospheric

particles under climate conditions as expected for the end of the century, and, to the best of our
knowledge, our study represents the first attempt to test for the combined effect of ocean warming
and acidification on these processes. Louis et al. (2018) have already shown from an abiotic dust
experiment that even an extreme ocean acidification scenario (~ -0.6 pH units) does not impact the
bioavailability of macro- and micro-nutrients ($NO_x$, DIP and DFe) for surface phytoplankton
communities in the oligotrophic Northwestern Mediterranean Sea, using the same dust analog and
simulated flux as used during our experiments. Similar results were presented by Mélançon et al.
(2016) regarding the release of DFe from dust in high-nutrient low-chlorophyll (HNLC) waters of





the Northeastern Pacific, following a mild ocean acidification scenario of -0.2 pH units. Our results
agree with these previous findings and further highlighted the absence of direct effect of ocean
warming (+3 °C) on the release of nutrients from atmospheric particles.

The differences in nutrient consumption dynamics between ambient and warmed/acidified

tanks were substantially dependent on the considered nutrient and investigated station. Regarding
$NO_x$, while no impacts of warming and acidification could be observed at stations TYR and ION
due to large variability between the duplicates (Table 4), larger $NO_x$ consumption rates were shown
under future climate conditions at the most productive station FAST as a consequence of strongly
enhanced biological stocks (see thereafter) and metabolic rates (Gazeau et al., in preparation, this
issue). The differences in DIP dynamics between the two dust-amended treatments were more
complex to interpret depending on the investigated station. A clear feature of our experiments is
that, in contrast to present day pH and temperature conditions, all the stock of DIP released from
dust was consumed at the end of the three experiments under future conditions. That being said, the
decreasing rates of DIP concentrations for that future conditions treatment differed depending on
the station (Table 4). While DIP dynamics were relatively similar between treatments at ION, a
clear effect of warming and acidification was shown at station TYR and FAST where the vast
majority of released DIP was consumed within 24 h ($\Delta$DIP = -1.3 and -1.1 to -1.5 nmol L$^{-1}$ h$^{-1}$ at
TYR and FAST, respectively). An interesting outcome at station TYR was that, despite the
important discrepancies observed for autotrophic stocks and metabolic rates between the duplicates
G1 and G2 (see section 4.2), a very similar dynamics was observed for DIP concentrations in these
tanks. As heterotrophic prokaryote biomass and production rates (Gazeau et al., in preparation, this
issue) did not differ between these duplicate tanks, this further highlights the clear dominance of
heterotrophic processes at this station, a dominance which was exacerbated by dust addition and
future climate conditions.



At station ION, DIP consumption rates were similar following dust addition under present
and future conditions. This results appears surprising as large impacts of warming and acidification
have been observed, especially for primary producers, as shown by almost doubled chlorophyll *a*
concentrations as compared to dust amended tanks (D). At this station, all autotrophic groups
benefited from ocean acidification and warming. *Synechococcus* and to a lesser extent
pico-eukaryotes appeared as the most impacted ones. Yet these differences of sensitivity among
autotrophs did not lead to detectable changes in the composition of the autotrophic assemblage as
compared to ambient conditions, with still a large dominance of nano-eukaryote carbon biomass at
the end of this experiment (62% in treatment G vs. 64% in treatment D). Very contrasted results
have been shown on the effect of ocean acidification on small autotrophic species (e.g. Dutkiewicz
et al., 2015) while there are increasing evidences that small phytoplankton species will be favored
in a warmer ocean (e.g. Chen et al., 2014; Daufresne et al., 2009; Morán et al., 2010). As mentioned
earlier, our experimental protocol was not conceived to discriminate temperature from pH effects,
however results concur with those of Maugendre et al. (2015) which further suggested temperature
over elevated $CO_2$ as the main driver of increased picophytoplankton abundance. As heterotrophic
prokaryotes were also positively impacted by future environmental conditions, the similarity of DIP
dynamics between ambient and future conditions suggests a tight coupling between the autotrophic
and heterotrophic compartments at this station. This is further evidenced by the absence of
differences detected over the relatively short time duration of our experiment on meso-zooplankton
abundance and carbon export efficiency (Gazeau et al., in preparation, this issue).
At FAST, similar to what was observed at station ION, all phytoplanktonic groups were
positively impacted by warming and acidification with the strongest changes detected for
*Synechococcus* as compared to ambient conditions. However, in contrast to station ION, all groups
reached maximal abundances (and carbon biomass) after 3 days of incubations, thereafter





drastically decreasing most likely as a consequence of DIP limitation (see above). It must be
stressed that this pattern could not be observed through pigment dynamics as no sampling was
performed for these analyses after 3 days of incubation. Also, in contrast to station ION, the
abundance of heterotrophic prokaryotes in the warmer and acidified treatment reached a maximum
after 2 days of incubations and then strongly decreased to reach levels observed in the control
treatment. This suggests that the heterotrophic compartment was the first to suffer from DIP
limitation and further highlights the dominance of the autotrophic compartment in terms of nutrient
consumption at this station. The differential dynamics of these two compartments under warmer and
acidified conditions most likely led to an excess production of organic matter that translated, for
instance, in higher dissolved organic carbon concentrations in this treatment (Gazeau et al., in
preparation, this issue). This excess production as compared to ambient conditions did not seem to
reach higher trophic levels as no clear differences in meso-zooplankton abundances were observed.
We fully acknowledge that the duration of our experiments was certainly too short to carefully
assess the proportion of newly formed organic matter consumed by meso-zooplanktonic species and
its effect on their abundances, yet group-specific variations were observed. Finally, it appeared that
at least part of this excess organic matter was exported to the bottom of the tanks as a higher carbon
export efficiency was observed at this station under warmer and acidified conditions (Gazeau et al.,
in preparation, this issue).


## 5. Conclusion

These experiments conducted during the PEACETIME cruise represent the first attempt to investigate the impacts of atmospheric deposition on surface plankton communities both under present and future environmental conditions. Despite few experimental issues that are discussed, the three experiments provided new insights on these potential impacts in the open Mediterranean Sea. Interestingly, the effect of dust deposition was highly different between the three investigated stations in the Tyrrhenian Sea, Ionian Sea and in the Algerian basin. As the initial conditions in the sampled surface seawater at the three stations were very similar in terms of nutrient availability and chlorophyll content, these differences rather seem to be a consequence of the initial metabolic states of the community (autotrophy vs. heterotrophy). In all three cases, nutrient addition from dust deposition did not strongly modify but rather exacerbated this initial state. Relative changes in main parameters presented in this manuscript and processes presented in Gazeau et al. (in preparation, this issue) as a consequence of dust addition under present and future environmental conditions are shown in Fig. 10, and compared to the compilation of published data for the Mediterranean Sea from Guieu and Ridame (2020). At station TYR, under conditions of a clear dominance of heterotrophs on the use of resources, dust addition drove the community into an even more heterotrophic state with no detectable effect on primary producers. At station ION, where the community was initially closer to metabolic balance, both heterotrophic and autotrophic compartments benefited from dust derived nutrients. At FAST, the most active station in terms of autotrophic production, addition of nutrients boosted both compartments but heterotrophic prokaryotes became quickly P-limited and overall larger effects were observed for phytoplankton. Ocean acidification and warming did not have any detectable impact on the release of nutrients from atmospheric particles. Furthermore, these external drivers did not drastically modify the





composition of the autotrophic assemblage with all groups benefiting from warmer and acidified
conditions. However, very large increases were observed both for autotrophic and heterotrophic
stocks and processes suggesting an exacerbation of effects from atmospheric dust deposition in the
future, rather than a change in the role of Mediterranean surface plankton community as a source or
a sink of $CO_2$ to or from the atmosphere.



# Data availability

All data and metadata will be made available at the French INSU/CNRS LEFE CYBER database
(scientific coordinator: Hervé Claustre; data manager, webmaster: Catherine Schmechtig).
INSU/CNRS LEFE CYBER (2020)

# Author contributions

FG and CG designed and supervised the study. FG, CG, CR and KD sampled seawater from the
experimental tanks during the experiments. JMG and GDL participated in the technical preparation
of the experimental system and all authors participated in sample analyses. FG, CR and CG wrote
the paper with contributions from all authors.

# Financial support

This study is a contribution to the PEACETIME project (http://peacetime-project.org), a joint
initiative of the MERMEX and ChArMEx components supported by CNRS-INSU, IFREMER,
CEA, and Météo-France as part of the programme MISTRALS coordinated by INSU.
PEACETIME was endorsed as a process study by GEOTRACES. PEACETIME cruise
(https://doi.org/10.17600/17000300).

# Acknowledgments

The authors thank the captain and the crew of the RV Pourquoi Pas ? for their professionalism and
their work at sea. We thank Julia Uitz, Céline Dimier and the SAPIGH HPLC analytical service at
Institut de la Mer de Villefranche (IMEV) for sampling and analysis of phytoplankton pigments,
John Dolan for microscopic countings as well as Lynne Macarez and the PIQv-platform of





EMBRC-France, a national Research Infrastructure supported by ANR, under the reference
ANR-10-INSB-02, for mesozooplankton analyses.





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



# Tables

Table 1. List of investigated parameters and processes during the three experiments at stations TYR, ION and FAST. Related manuscripts are indicated. $pH_T$: pH on the total scale, $A_T$: total alkalinity, $^{13}C$-$C_T$: $^{13}C$ signature of dissolved inorganic carbon, $NO_x$: nitrate + nitrite, DIP: dissolved inorganic phosphorus, $Si(OH)_4$: silicate, DFe: dissolved iron, DAl: dissolved aluminium, Th-REE-Pa: Thorium (230Th and 232Th), Rare Earth elements and Protactinium (231Pa), POC: particulate organic carbon, DOC: dissolved organic carbon, $^{13}C$-DOC: $^{13}C$ signature of dissolved organic carbon, TEP: transparent exopolymeric particles, NCP/CR: net community production and community respiration (oxygen based), $^{14}C$-PP: primary production based on $^{14}C$ incorporation.



| Sampling time | T-1 | T0 | T1 | T2 | T3 | T4 | T5 | T6 / T7 | Related manuscript |
| --- | --- | --- | --- | --- | --- | --- | --- | --- | --- |
| | Filling tanks | Before seeding, after warming / acidification | +1 h | +6 h | +12 h | +24 h | +48 h | +72 h /+96 h | |
| Temperature | | | Continuous | | | | | | This manuscript |
| Irradiance | | | Continuous | | | | | | This manuscript |
| **Carbonate chemistry** | | | | | | | | | |
| pH$_T$ | ▓ | ▓ | | | | ▓ | ▓ | ▓ | This manuscript |
| $A_T$ | ▓ | ▓ | | | | ▓ | ▓ | ▓ | This manuscript |
| δ$^{13}$C-$C_T$ | ▓ | ▓ | | | ▓ | | | | Gazeau et al. (in preparation) |
| **Macro-nutrients** | | | | | | | | | |
| NO$_x$ | ▓ | ▓ | ▓ | ▓ | ▓ | ▓ | ▓ | ▓ | This manuscript |
| DIP | ▓ | ▓ | ▓ | ▓ | ▓ | ▓ | ▓ | ▓ | This manuscript |
| Si(OH)$_4$ | ▓ | ▓ | ▓ | ▓ | ▓ | ▓ | ▓ | ▓ | This manuscript |
| **Micro-nutrients** | | | | | | | | | |
| DFe | ▓ | ▓ | | | | ▓ | | ▓ | Roy-Barman et al. (in preparation) |

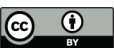

| Parameter | Reference |
|---|---|
| DAl | Roy-Barman et al. (in preparation) |
| Th-REE-Pa | Roy-Barman et al. (in preparation) |
| **Biological stocks** | |
| Pigments | This manuscript |
| Flow cytometry | This manuscript |
| Microscopy | This manuscript |
| Diazotroph abundance | Ridame et al. (in preparation) |
| Virus abundance | Dinasquet et al. (in preparation) |
| Meta-transcriptomics | Dinasquet et al. (in preparation) |
| Bacterial diversity | Dinasquet et al. (in preparation) |
| Micro-eukaryote diversity | Dinasquet et al. (in preparation) |
| Meso-zooplankton | This manuscript |
| POC (incl. $\delta^{13}C$) | Gazeau et al. (in preparation) |
| POC sediment traps | Gazeau et al. (in preparation) |
| DOC | Gazeau et al. (in preparation) |
| $^{13}C$-DOC | Gazeau et al. (in preparation) |
| TEP | Gazeau et al. (in preparation) |



| Processes | | Reference |
|---|---|---|
| Amino acids | | Gazeau et al. (in preparation) |
| Carbohydrates | | Gazeau et al. (in preparation) |
| NCP/CR | | Gazeau et al. (in preparation) |
| $^{14}$C-PP | | Gazeau et al. (in preparation) |
| Heterotrophic production | | Gazeau et al. (in preparation) |
| Ectoenzymatic activity | | Gazeau et al. (in preparation) |
| $N_2$ fixation | | Ridame et al. (in preparation) |
| $^{13}CO_2$-fixation | | Ridame et al. (in preparation) / Gazeau et al. (in preparation) |
| Virus production, lysogeny | | Dinasquet et al. (in preparation) |





Table 2. Initial conditions as measured while filling the tanks (initial conditions in pumped surface
water; sampling time: t-12h). $pH_T$: pH on the total scale, $NO_x$: nitrate + nitrite, $NH_4$: ammonium,
DIP: dissolved inorganic phosphorus, $Si(OH)_4$: silicate, TChl$a$: total chlorophyll $a$, HNF:
heterotrophic nanoflagellates. The three most important pigments in terms of concentration are also
presented (19'-hexanoyloxyfucoxanthin, Zeaxanthin and Divinyl Chlorophyll $a$). Biomasses of the
different groups analyzed through flow cytometry were estimated based on conversion equations
and/or factors found in the literature (see section 2.3). Autotrophic biomass was, as a first
approximation, estimated only based on flow cytometry data and therefore corresponds to the
fraction < 20 µm. Heterotrophic biomass was estimated as the sum of heterotrophic prokaryote and
HNF biomasses (see section 2.3). Values below detection limits are indicated as < dl.



| Sampling station | | TYR | ION | FAST |
|---|---|---|---|---|
| | Coordinates (decimal) | 39.34 N, 12.60 E | 35.49 N, 19.78 E | 37.95 N, 2.90 N |
| | Bottom depth (m) | 3395 | 3054 | 2775 |
| | Day and time of sampling (local time) | 17/05/2017 17:00 | 25/05/2017 17:00 | 02/06/2017 21:00 |
| | Temperature (°C) | 20.6 | 21.2 | 21.5 |
| | Salinity | 37.96 | 39.02 | 37.07 |
| Carbonate chemistry | $pH_T$ | 8.04 | 8.07 | 8.03 |
| | Total alkalinity ($\mu$mol kg$^{-1}$) | 2529 | 2627 | 2443 |
| Nutrients | $NO_x$ (nmol L$^{-1}$) | 14.0 | 18.0 | 59.0 |
| | $NH_4^+$ ($\mu$mol L$^{-1}$) | 0.045 | 0.022 | <dl |
| | DIP (nmol L$^{-1}$) | 17.1 | 6.5 | 12.9 |
| | $Si(OH)_4$ ($\mu$mol L$^{-1}$) | 1.0 | 0.96 | 0.64 |
| | $NO_x$/DIP (molar ratio) | 0.8 | 2.5 | 4.6 |
| Pigments | TChl$a$ ($\mu$g L$^{-1}$) | 0.063 | 0.066 | 0.072 |
| | 19'-hexanoyloxyfucoxanthin ($\mu$g L$^{-1}$) | 0.017 | 0.021 | 0.016 |
| | Zeaxanthin ($\mu$g L$^{-1}$) | 0.009 | 0.006 | 0.036 |
| | Divinyl Chlorophyll $a$ ($\mu$g L$^{-1}$) | ~0 | 0 | 0.014 |




| | | | | |
|---|---|---|---|---|
| Flow cytometry | Pico-eukaryotes (abundance in cell mL⁻¹; biomass in µg C L⁻¹) | 347.8; 0.5 | 239.9; 0.4 | 701.0; 1.0 |
| | Nano-eukaryotes (abundance in cell mL⁻¹; biomass in µg C L⁻¹) | 150.5; 3.9 | 188.8; 4.8 | 196.6; 5.0 |
| | *Synechococcus* (abundance in cell mL⁻¹; biomass in µg C L⁻¹) | 4972; 1.2 | 3037; 0.8 | 6406; 1.6 |
| | Autotrophic biomass (µg C L⁻¹) | 5.6 | 6.0 | 7.7 |
| | Heterotrophic prokaryotes abundance (x 10⁵ cell mL⁻¹) | 4.79 | 2.14 | 6.15 |
| | HNF (abundance in cell mL⁻¹) | 110.1 | 53.6 | 126.2 |
| | Heterotrophic biomass (µg C L⁻¹) | 9.9 | 4.5 | 12.7 |
| Microscopy | Pennate diatoms (abundance in cell L⁻¹) | 140 | 520 | 880 |
| | Centric diatoms (abundance in cell L⁻¹) | 200 | 380 | 580 |
| | Dinoflagellates (abundance in cell L⁻¹) | 2770 | 3000 | 3410 |
| | Autotrophic flagellates (abundance in cell L⁻¹) | 0 | 60 | 650 |
| | Ciliates (abundance in cell L⁻¹) | 270 | 380 | 770 |

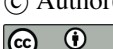


Table 3. Maximum input of nitrate + nitrite ($NO_x$) and dissolved inorganic phosphorus (DIP)
released from Saharan dust in tanks D and G as observed from the two discrete samplings
performed over the first 6 h after seeding. The estimated maximal percentage of dissolution is also
presented (see section 2.3 for details on the calculations).

|  |  | $NO_x$ | | | | DIP | | | |
|---|---|---|---|---|---|---|---|---|---|
|  |  | D1 | D2 | G1 | G2 | D1 | D2 | G1 | G2 |
| Maximum input |  | $\mu$mol L$^{-1}$ | | | | nmol L$^{-1}$ | | | |
|  | TYR | 11.0 | 11.1 | 11.1 | 11.0 | 24.6 | 20.4 | 24.6 | 23.9 |
|  | ION | 11.2 | 11.6 | 11.2 | 11.3 | 23.3 | 22.0 | 19.6 | 22.9 |
|  | FAST | 11.3 | 11.1 | 11.1 | 11.2 | 30.8 | 31.3 | 36.9 | 29.8 |
| Percentage of dissolution (%) |  |  |  |  |  |  |  |  |  |
|  | TYR | 95 | 96 | 95 | 94 | 12 | 10 | 12 | 11 |
|  | ION | 96 | 99 | 96 | 97 | 11 | 10 | 9 | 11 |
|  | FAST | 97 | 97 | 95 | 97 | 15 | 15 | 17 | 14 |






Table 4. Removal rate of nitrate + nitrite ($NO_x$) and dissolved inorganic phosphorus (DIP) in tanks
D and G during the three experiments (TYR, ION and FAST). For $NO_x$, decreasing rates were
estimated based on linear regressions between maximal concentrations (i.e. after dust enrichment, at
t1h or t6h) and final concentrations (t72 h for TYR and ION and t96h for FAST). For DIP,
decreasing rates were estimated based on linear regressions between maximal concentrations (i.e.
after dust enrichment at t1h or t6h) and concentrations measured at sampling times after which a
stabilization was observed. This sampling time is shown in parentheses. All rates are expressed in
nmol $L^{-1}$ $h^{-1}$.

| | $NO_x$ | | | DIP | | |
|---|---|---|---|---|---|---|
| | TYR | ION | FAST | TYR | ION | FAST |
| D1 | -6.5 | -8.6 | -14.3 | -0.4 (t72h) | -0.5 (t48h) | -0.2 (t96h) |
| D2 | -1.0 | -8.6 | -13.5 | -0.3 (t72h) | -0.8 (t24h) | -0.2 (t96h) |
| G1 | -6.7 | -13.1 | -21.6 | -1.3 (t24h) | -0.8 (t24h) | -1.5 (t24h) |
| G2 | -0.8 | -1.6 | -25.2 | -1.3 (t24h) | -1.6 (t24h) | -1.1 (t24h) |




Table 5. Maximum relative changes in tanks D and G as compared to controls (average between C1
and C2), expressed as a %, for the three experiments (TYR, ION and FAST). The sampling time at
which these maximum relative changes were observed is shown in brackets. Tchl$a$ refers to the
concentration of total chlorophyll $a$ and $B_{micro}$ to the biomass proxy of micro-phytoplankton (sum of
Fucoxanthin and Peridinin, see Material and Methods) based on high performance liquid
chromatography (HPLC). HP and HNF refer to heterotrophic prokaryote and heterotrophic
nanoflagellate abundances, respectively, as measured by flow cytometry.





| Experiment | Tank | HPLC | | Flow cytometry | | | | |
| --- | --- | --- | --- | --- | --- | --- | --- | --- |
| | | TChl*a* | B$_{micro}$ | Pico-eukaryotes | Nano-eukaryotes | *Synechococcus* | HP | HNF |
| TYR | D1 | -35 (t24h) | -33 (t12h) | -75 (t72h) | -80 (t1h) | -71 (t48h) | 68 (t72h) | 352 (t72h) |
| TYR | D2 | -38 (t12h) | -39 (t24h) | -75 (t72h) | -80 (t1h) | -72 (t48h) | 53 (t72h) | 100 (t72h) |
| TYR | G1 | 60 (t72h) | 52 (t72h) | -75 (t1h) | 89 (t72h) | 76 (t72h) | 67 (t72h) | 1095 (t72h) |
| TYR | G2 | 359 (t72h) | 392 (t72h) | 323 (t72h) | 119 (t72h) | 700 (t72h) | 68 (t48h) | 298 (t72h) |
| ION | D1 | 183 (t72h) | 157 (t72h) | 126 (t72h) | 89 (t72h) | 317 (t72h) | 128 (t72h) | 44 (t72h) |
| ION | D2 | 109 (t72h) | 156 (t72h) | 117 (t72h) | -59 (t1h) | 390 (t72h) | 133 (t72h) | 27 (t72h) |
| ION | G1 | 399 (t72h) | 454 (t72h) | 458 (t72h) | 256 (t72h) | 805 (t72h) | 176 (t72h) | 175 (t72h) |
| ION | G2 | 426 (t72h) | 612 (t72h) | 510 (t72h) | 292 (t72h) | 1425 (t72h) | 161 (t72h) | 129 (t72h) |
| FAST | D1 | 318 (t96h) | 356 (t96h) | 113 (t96h) | 208 (t72h) | 348 (t96h) | 27 (t96h) | -38 (t96h) |
| FAST | D2 | 237 (t96h) | 322 (t96h) | 91 (t96h) | 219 (t72h) | 197 (t96h) | 40 (t48h) | -49 (t96h) |
| FAST | G1 | 399 (t96h) | 415 (t96h) | 198 (t72h) | 274 (t72h) | 357 (t48h) | 61 (t48h) | 243 (t24h) |
| FAST | G2 | 395 (t96h) | 421 (t96h) | 129 (t72h) | 202 (t96h) | 344 (t48h) | 67 (t48h) | 74 (t24h) |





## Figure captions

Fig. 1. Map showing the sampling stations in the Mediterranean Sea along the transect performed onboard the R/V "Pourquoi Pas ?" during the PEACETIME cruise.

Fig. 2. Scheme of an experimental tank (climate reactor).

Fig. 3. Proportion of the different pigments, as measured by high performance liquid chromatography (HPLC) in pumped surface seawater for the three experiments (t-12h).

Fig. 4. Continuous measurements of temperature and irradiance level (PAR) in the six tanks during the three experiments. The dashed vertical line indicates the time of dust seeding (after t0).

Fig. 5. pH on the total scale ($pH_T$) and total alkalinity ($A_T$) measured in the six tanks during the three experiments. The dashed vertical line indicates the time of dust seeding (after t0). Error bars correspond to the standard deviation based on analytical triplicates.

Fig. 6. Nutrients (nitrate + nitrite: $NO_x$, dissolved inorganic phosphorus: DIP, silicate: $Si(OH)_4$ as well as the molar ratio between $NO_x$ and DIP, measured in the six tanks during the three experiments. The dashed vertical line indicates the time of seeding (after t0).

Fig. 7. Concentrations of total chlorophyll *a* and major pigments, measured by high performance liquid chromatography (HPLC), in the six tanks during the three experiments. The dashed vertical line indicates the time of seeding (after t0).

Fig. 8. Abundance of pico-eukaryotes, nano-eukaryotes, *Synechococcus,* heterotrophic prokaryotes (HP), and heterotrophic nano-flagellates (HNF), measured by flow cytometry, in the six tanks during the three experiments. The evolution of autotrophic biomass (see Material and Methods for



details on the calculation) is also shown. The dashed vertical line indicates the time of seeding (after
t0).
Fig. 9. Abundances of meso-zooplankton species as measured at the end of each experiment.
Fig. 10. Maximal relative change (%) of main biological stocks (TCHl*a*: total chlorophyll *a*, HP:
heterotrophic prokaryotes) and processes (BP: bacterial production; PP: [14]C-based primary
production; see Gazeau et al., in preparation, this issue; BR: bacterial respiration (no data from this
study); and $N_2$ fixation, see Ridame et al., in preparation, this issue) as obtained during the present
study at the 3 stations (TYR, ION and FAST) under ambient conditions of pH and temperature
(open red squares) and future conditions (full green squares). Squares are delimited by the range of
responses observed among the duplicates for each treatment. The dotted green squares for station
TYR denote the large variability observed between duplicates for some parameters and processes
that prevented drawing solid conclusions. Box-plots represent the distribution of responses
observed from studies conducted in the Mediterranean Sea, as compiled by Guieu and Ridame

1267 (2020).




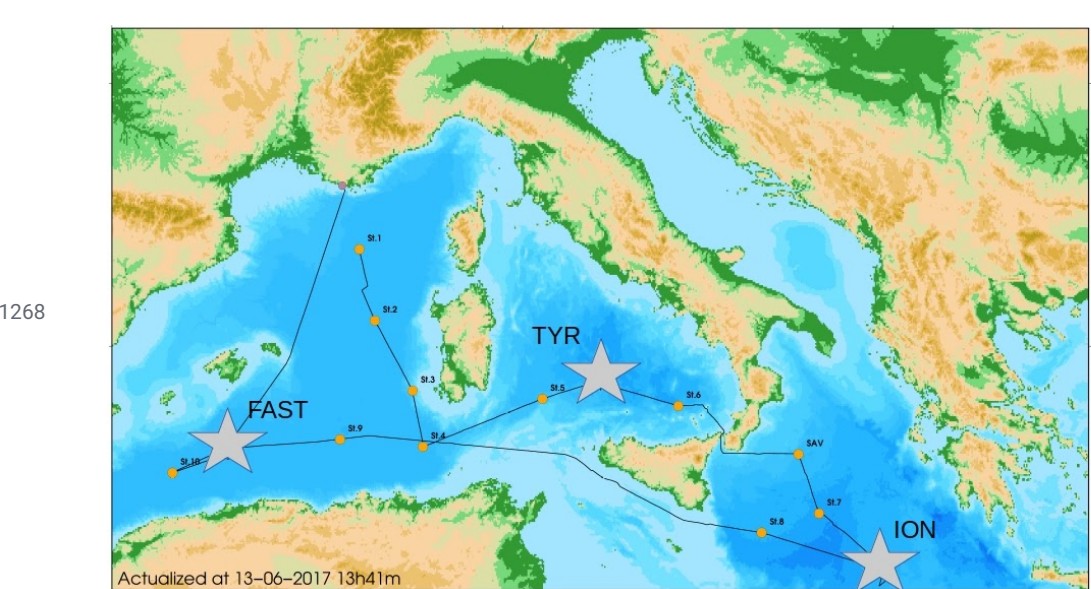

Fig. 1.




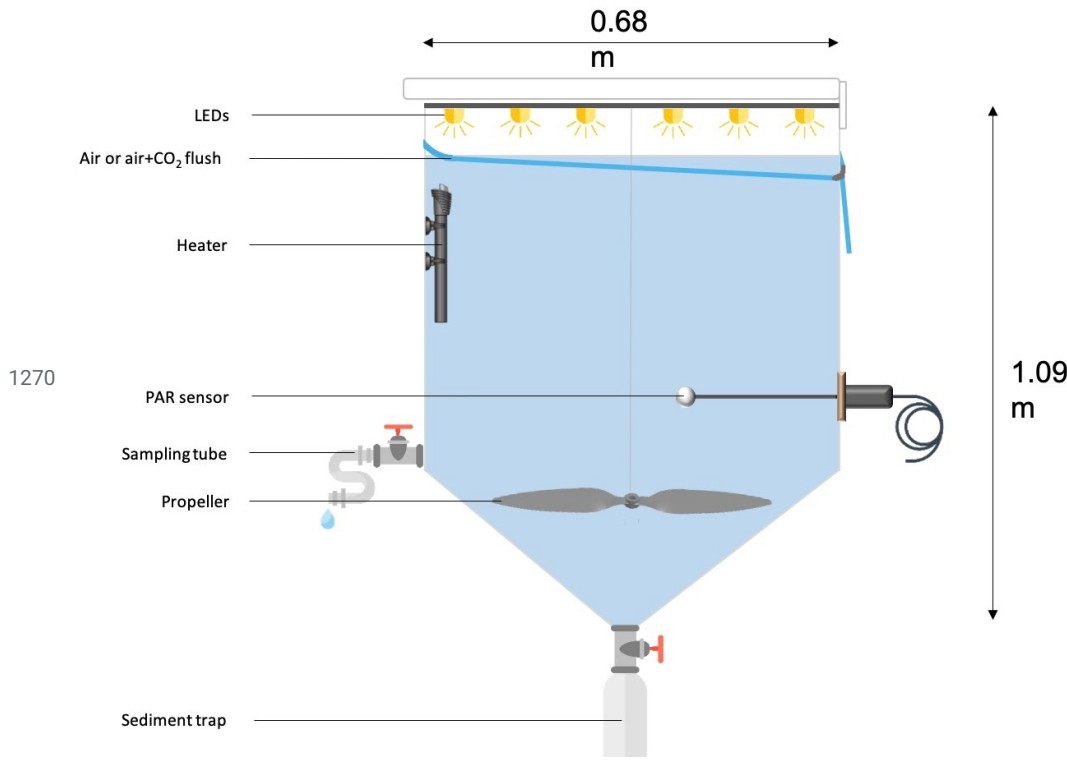

Fig. 2.




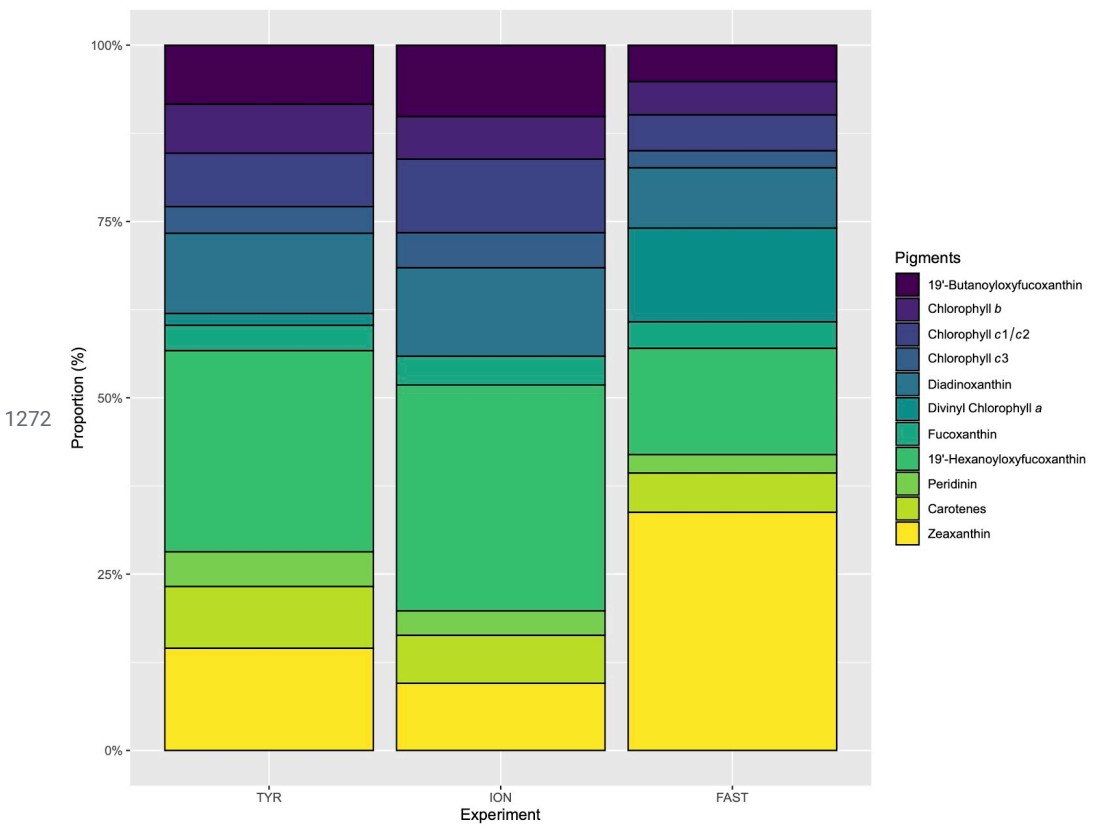


Fig. 3.



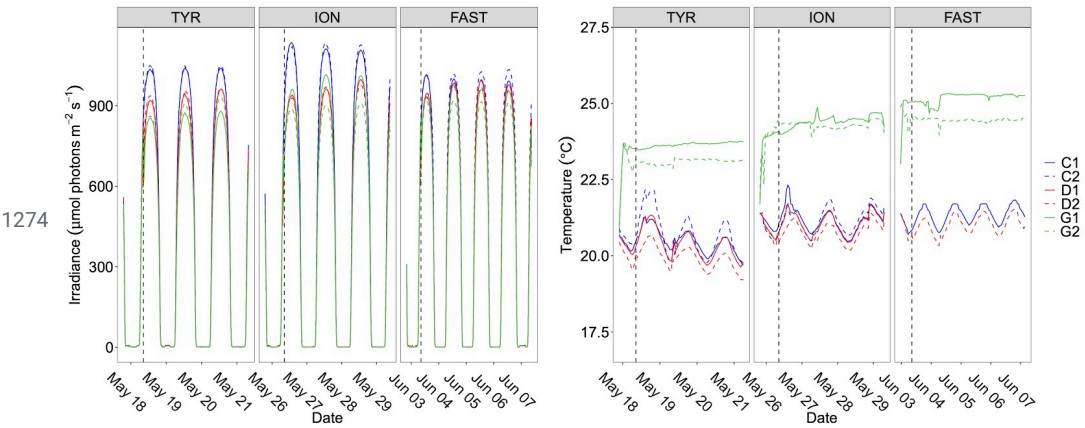


Fig. 4.




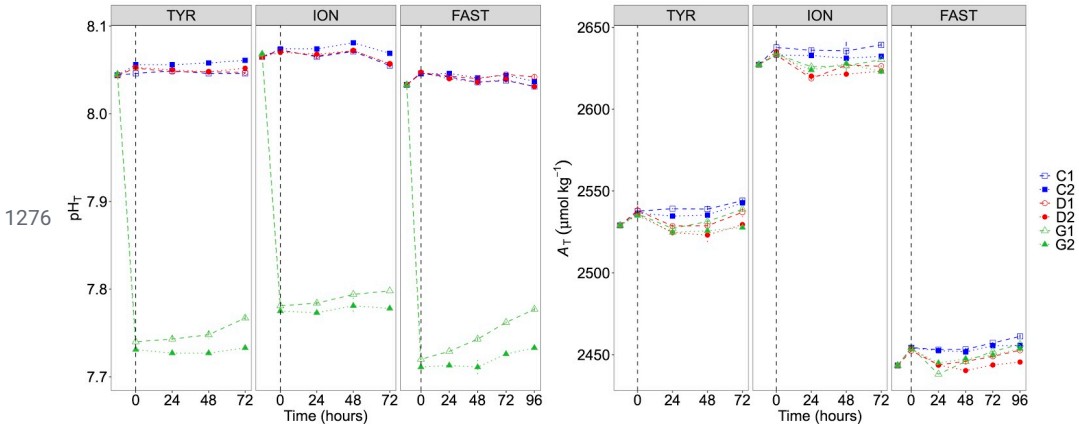

Fig. 5.



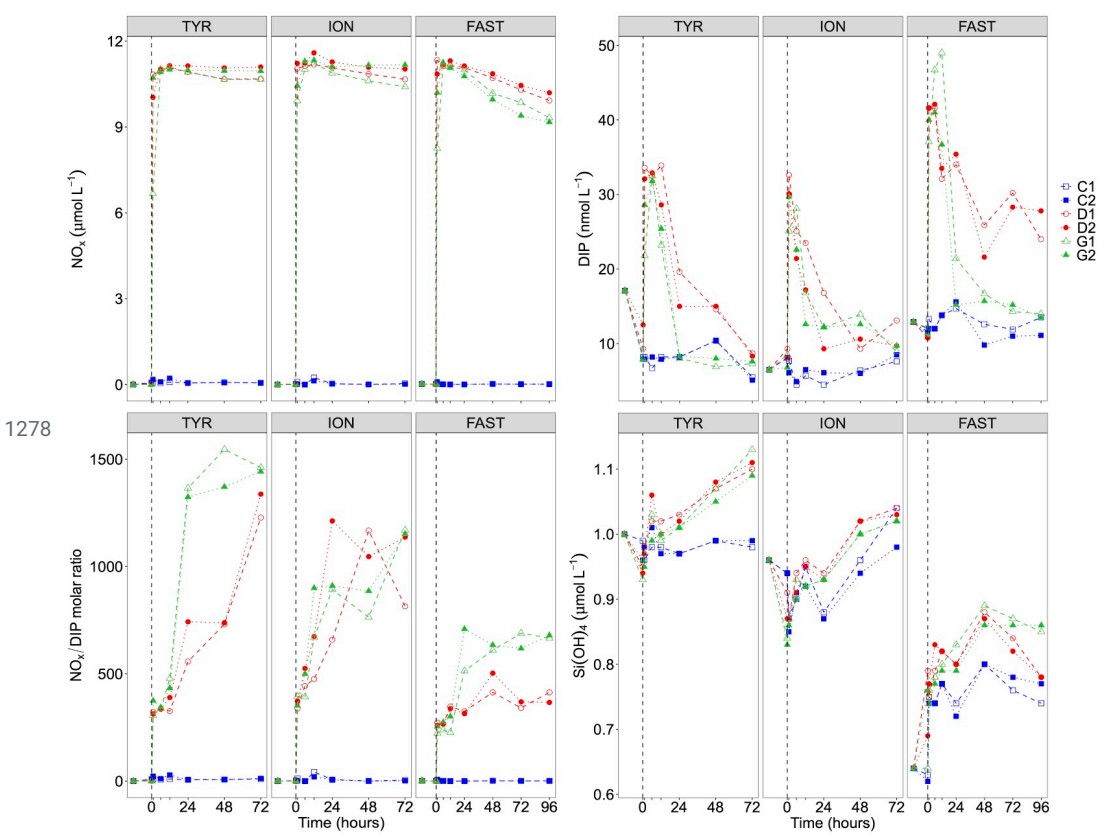


Fig. 6.





Fig. 7.




Fig. 8.



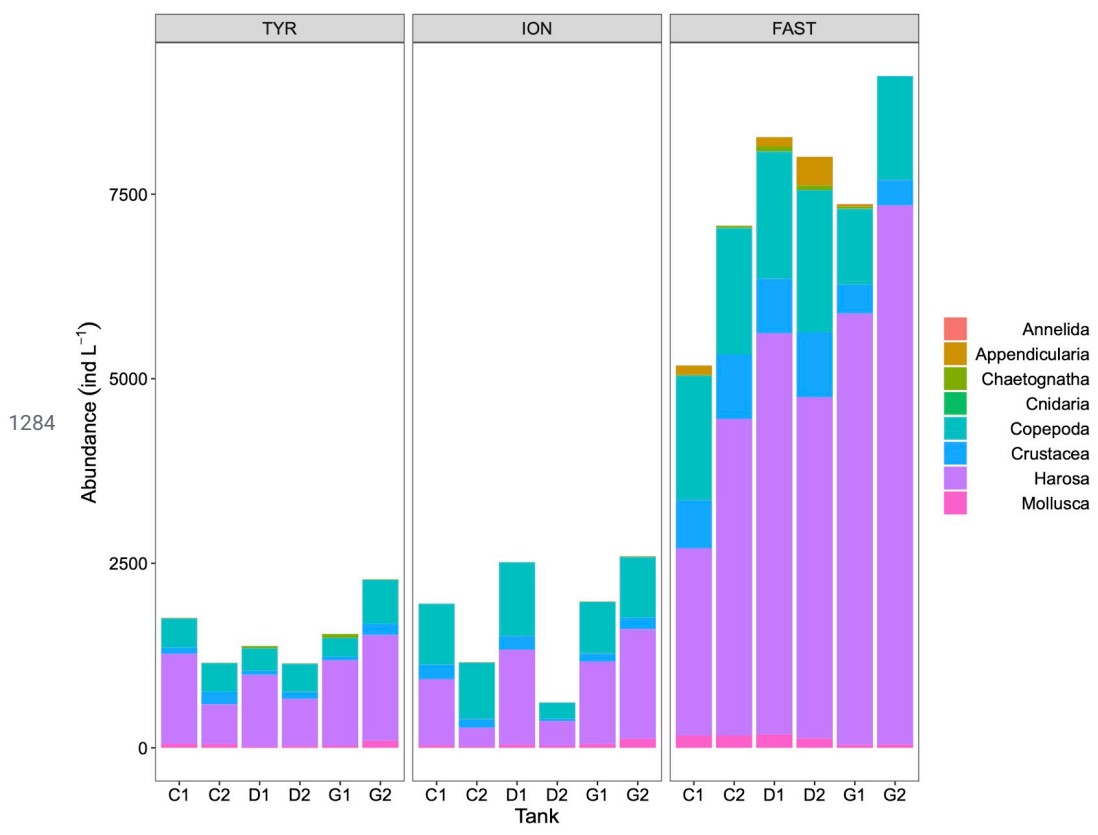

Fig. 9.



Fig. 10.