# Peer review of "communities under present and future conditions of pH and"

_Biogeosciences, 2020_

## Referee Comment (RC1) · Anonymous Referee #1 · 22 Sep 2020

General comments: I consider the contribution by Gazeau et al., to be an extremely interesting and relevant study attempting at addressing important questions regarding the role of atmospheric dust deposition as an alternative source of bioavailable nutrients for ocean productivity in the warmer and more acidified ocean projected for the future.

The design of the experiment is excellent and very well-conceived. I particularly like the fact that it was run at 3 distinct environmental settings along a W-E gradient, providing an enormous potential for exploring changes/differences related to spatial/regional

variability along the Mediterranean Sea.

This important study highlights how complex and diversified can be the responses of distinct biological groups to environmental variability, in this case, dust-born nutrient addition and ocean warming and acidification. Results from this work stress the urgent need for monitoring modern marine ecosystems and their spatiotemporal relationship to environmental parameters in order to better understand how they are/will respond to climate change. The study submitted by Gazeau et al., clearly has the potential to provide a robust discussion on these important and timely topics.

However, in its present form, the manuscript is not ready yet for publication, whereas a major revision and structure reorganization is recommended. Overall, I find that the paper lacks focus, and the interpretation of the results could benefit from some more maturation. The discussion, in particular, is hard to follow, bringing difficulties in getting a clear overview on how the different abiotic and biological parameters varied throughout time and space, and how the several hypotheses proposed for the observed trends are entangled/linked. The ideas could be better organized and entangled: whether you decide to compare each variable along different treatments/sites, or you decide to compare different sites in relation to all the variables (i.e. you provide a "general and integrated picture" for each site and there compare the three), whatever logic you use, please stick with it along the entire ms. for consistency. Overall, the main structure of the discussion could be organized in terms of the main differences abiotic and biotic observed along W-E gradient, and how such gradients in the initial/original in situ conditions are likely to modulate/be modulated by dust deposition and ocean warming and acidification.

Given the larger number of parameters and biological groups, for which you used different sub-sampling and analytical approaches, the set-up and procedure should be described more clearly, maybe adding an experiment schematic timeline.

The manuscript could also benefit from a review in terms of written English.

Specific comments:

Section 4.1 The discussion would greatly benefit from being shortened and written in a more focused manner, instead of often repeating the results and providing too many unnecessary details that prevent the reader for getting the "general picture".

You should (a) provide a clear and focused global picture of the initial conditions along the FAST-TYR-ION transect, while (b) directly addressing potential causes/explanations for the most important/relevant patterns and observations described in the results.

Section 4.2 I understand that the authors are discussing the strengths and limitations of their experiment, but I am not sure if they need an entire section for this, and especially not before discussing the results. What matters the most is the "story" they can tell from their spatiotemporal experiment, and what kind of insights on the effects of dust deposition and ocean warming/acidification they can extract from this story.

Section 4.3 Overall, this section could also benefit of more focus. The arguments should be better entangled while comparing a) different treatments, b) different sites, and c) with previous studies. Some parts are very confusing and difficult to follow.

It is not easy for the reader to reach the end of this section having a clear picture on which groups benefited the most from dust, where and why, and how that translated into the whole ocean trophic chain and export of organic matter. Although this the center-theme of the paper, it is not written in a very clear/straightforward manner. The authors should be able to discuss the difference between dust deposition along a W-E gradient along which the abiotic conditions vary. Potential impacts for the biological carbon pump could also be discussed.

Section 4.4 This section has similar issues than the previous sections, concerning its somewhat confusing and poorly organized writing/discussion. Furthermore, I also find it very similar to what is presented in the results. Following a clear and to-the-point

presentation of the most striking differences between D and G treatments, the authors should take the chance to really discuss/reflect on the meaning of such differences in the light of ongoing ocean warming and acidification. The section revolves too much about details and differences amongst different sites/treatments but does not provide an actual "big picture" of the experiment nor a reflection on what the latter means.

Conclusion: I like this chapter because it is written following a more systematic, logic and to-the-point order. The discussion chapters would greatly benefit from being re-written in a similar manner. In order to do that I would suggest the following approach:

- present a clear discussion of the initial conditions: you need to provide a clear abiotic and biotic spatial picture of your experiment. The environmental background is often not even mentioned. This is a pity since you have data for comparing three distinct environmental settings along a W-E gradient. A good discussion on the differences amongst different stations following dust/warming/acidification should take into account the environmental differences in their initial conditions.

- present a clear discussion on the effects of dust compared to the control, taking into account the differences in the initial conditions amongst the three sites. Differences between duplicates in each station and the potential causes for them should be referred after. Take the chance to discuss important questions such as: will increasing dust deposition increase ocean productivity, CO2 fixation and CO2 sequestration? If so, which groups are likely to contribute the most and where? Are dust effects likely to influence the entire marine trophic chain or not? Etc.

- present a clear discussion on the effects of dust in the projected future ocean compared to the modern ocean. Are the differences in the initial conditions a relevant factor for the observed differences and/or similarities? Can you say anything about ocean productivity and the biological pump of the future based on your observations? Will dust help to counterbalance the nutrient-depletion from ocean warming? What can you say about the effects of acidification on calcifying groups and does that matter?

Etc.

Abstract Line 26: caused by climate-driven enhanced stratification

Line 29: the potential impact (...) was investigated

Line 36: maybe "analysis" instead of "processes"?

Line 38: impacts of dust seeding with and without addressing the effects of climate change

Line 44: I would provide more information regarding such differences and regarding the conditions at the three studied locations.

I would include more detailed information on which "biological stocks" you refer to, and which approach/method you used to explore the different biological groups.

Introduction Line 59: Please specify whether this Chl-a concentration refers to the surface, DCM or an average for the entire photic zone

Line 65: can you explain/frame what are dust emissions from anthropogenic origin?

Line 76: how did these authors justify this lack of agreement?

Line 62-76: I would break this paragraph in two.

Line 78: in that part of the basin

Line 78-80: present a higher impact as a source of bioavailable fertilizing nutrients compared to dry deposition, as confirmed on...

Line 79-80: which parameters and processes were those?

Line 81: using both micro- and mesocosms

Line 83: while also modifying...

Line 84: In addition, besides... also modified the...

Line 87: can you explain what you mean with "budgets"?

Line 89: ... heterotrophic biological behavior in these oligotrophic waters

Line 94: how can dust aggregate and ballast OM that is dissolved?

Line 101: ...nutrient cycling in the open ocean is being and will continue to be perturbed in the next decades, very likely to result in regionally variable impacts...

Line 104: ... due to thermal stratification-linked reduction of nutrient supply... (...) to which an increasing role of atmospheric dust deposition might contribute to compensate as an alternative source....

Line 106: Whether or not plankton communities will respond differently to dust deposition under more acid and stratified conditions due to ongoing ocean warming and acidification, both globally and regionally in the Mediterranean Sea (refs.) is still largely unknown.

Line 111: this sentence sounds strange; I would never expect that severe nutrient limitation and enhanced warming would ever lead to increase PP... or did you mean heterotrophic production?

Line 119: what about calcifying plankton, which have an important goal for both the organic and inorganic carbon pumps?

Line 144: this "and/or" is not very clear....

Line 145-147: do you mean other papers which were also submitted to this BG issue? "companion paper" does not sound well...

2. Material and Methods 2.1. General setup

Line 150: Six experimental tanks were installed in a T-controlled container, allowing to finely control the irradiance spectrum and intensity to fully reproduce a future scenario of enhanced ocean acidification and warming conditions.

Line 153: were trace-metal free? Or ARE trace-metal free?

Line 180: how long did it take between sub-sampling for initial conditions and dust seeding? Can you trust that the conditions inside the tanks did not change during this interval (nutrient-consumption, grazing, OM degradation, etc)?

Line 188: "for many parameters" sounds a bit vague..

Line 205: I would only show the parameters that are used in the results/discussion of this paper.

Line 205: are 3 days enough? What were your criteria to define the time-interval for the experiments? Which biological groups were you focusing on? It would be interesting to monitor the ecological succession of distinct plankton groups following the addition of dust.

Line 210: what do you mean by filtered "online"?

2.2. Analytical methods 2.2.2. Nutrients

Line 239: again the expression "filtered online".

2.2.4. Flow cytometry

Line 269: did you get any to details on the taxonomy of autotrophic nannoplankton?

2.2.5. Micro-phytoplankton and -heterotrophs

Line 284: If the filling of the tanks took 2h, and if you have sampled such surface seawater while filling the tanks... why do you call this sampling time "t-12h"? The temporal sequence of the (1) filling, (2) filled-tanks, (3) dust seeding, (4) sub-sampling moments, and (5) which parameters you sub-sampled and when, is not fully clear to me...

2.2.6. Mesozooplankton Line 290: why was FAST longer? When exactly was the dust added to the tanks? T-?

2.3. Computation What do you mean by "Computation"? Line 303: The maximum percentage of dust-born dissolved N and P... Line 311: How many times did you sub-sample for nutrients over those 6h following dust seeding?

Line 312: not sure I understand the way you determine the CONCdust...

Line 325: ... unites was done ...

Line 331: what about Chl-a?

3. Results 3.1. Initial conditions

Line 335: the part: "when pumping seawater for the experiments" sounds repetition of the first sentence. Just directly start describing what were the initial conditions.

Line 365: maybe this section should be titled "Experimental conditions at t0" Line 366: in the control tanks Lines 388-391: this part is not fully clear. Please specify to which tanks/treatments you added 13C and dust, and how was the sequence (which did you add first?)

Line 425: At TYR, while concentrations remained stable in control tanks, ...

Line 437: ... temporal dynamics showed very different patterns amongst the three studied stations.

Line 449: I would replace "at the exception of" by "with the exception of" throughout the ms.

4. Discussion 4.1. Initial conditions

Line 499: typical stratified

Line 500: please specify which period of the year you are referring to.

Line 501: ... were even lower than the ones.... during spring

Lines 514 and 515: please replace "whether.... or" by "both ".... and"

Lines 498-516: Maybe you don't need to provide such a long description given that your nutrient values are not directly comparable to those from the referred studies. It might be enough if you simply argue that in spite of not being possible to directly compare them, your results suggest highly oligotrophic conditions at the start of the experiment, in line with previous studies in the region and neighbor areas, and as typically expected for the open Mediterranean sea during spring. Basically, what you eventually say in lines 516-517.

Line 520: what do you mean by "enrichment experiments"? you mean experiments dealing with enrichment in DIP, nitrates, etc? And combination of what? Please explain.

Lines 520-527: it is not clear whether you are referring to experiments that you did or whether you are referring to studies by other authors (refs?). If I understood correctly, you are saying that productivity in the initial conditions were marked by N- and P- limitation but not by dFe. . .?

Line 528: Total concentrations of Chl-a . . .. were in line with low Chl-a levels found. . ... both driven from remote sensing satellite images . . . and from in situ measurements provided in a database from. . ..

Line 531: . . . low Chl-a concentrations around. . .

Lines 533-538: this part sounds more like a description/repetition of the results rather than its actual discussion.

Line 540: . . . a fingerprint of LNLC areas in general, and of surface Mediterranean waters during spring in particular.

Line 554: you should only cite works "submitted" or refer to them as "unpublished data" or "person. communication"

This section could benefit from being shortened and much more focused. As it is, the arguments are not easy to follow (often presented in "circles") making it hard to get a clear picture of what were in fact the initial conditions in the studied sites. For more

details, see the comments bellow.

Line 568: Not sure if the title fits the content of this section; I would maybe call it "Environmental conditions during the experiment" or something.

Line 569:. . . have been successfully validated in previous studies. . . to investigate the biological effects from the input of. . .. and resulting export of organic matter. . . under pre-defined close-to-abiotic conditions,

Line 574: which ones ate "these"? please clarify.

Lines 574-575: . . . no control of atmospheric $CO_2$ was, however, performed, resulting in a rapid increase of the pH levels in the acidified filtered seawater due to $CO_2$ outgassing. . .

Line 576: In order to avoid this, we improved. . .. reactors). This allowed us to significantly reduce $CO_2$ outgassing while maintaining pH levels. . ..

Line 580: Still, as illustrated in Fig. 5, the regulation of atmospheric $CO_2$ was. . . compared to G1, possibly due to a potential. . ..

Line 583: I know what you mean here, but this should be said in a different way and properly supported by the relevant results.

Line 588-592: too much descriptive. Please be more concise and directly address the potential causes for the observed discrepancies.

If the lids are the same, why would the intensity vary? And what do you mean by "PARs sensors sensitivity"? Are you referring to enhanced levels of turbidity when you refer to "amount of particles"? And why only for treatments G?

Line 597: you had the same type of unforeseen variability in terms of temperature? Only for G, or to all the treatments? This part should be better explained and discussed to what extend it might lead to misleading conclusions.

Line 599-600: "... but the new results are not yet available" (I am not sure what you want to state in the last sentence....)

Line 601: this is already well established at this point, there is no need to repeat this.

Line 605-612: Some of these sentences are too long. I am also not sure if this "introspection" about the criteria/available conditions for the design of your experiment is really relevant here. I would focus on providing a good discussion on the conditions prevailing during the experiment, how much they mimicked the settings that you wanted to mimic, how well the experiment conditions were in line with previous similar studies, and what is it that is new. You can refer to the main difficulties and challenges during its implementation, but always having in mind whether the latter allowed you or not to realistically (enough?) simulate/test your hypothesis. You should stick to discuss the quality of your experiment based on your observations/results. Anything other than that creates dispersion and makes the reading more difficult.

613-616: I understand this! However, this highlights the importance of providing a clear, to-the-point description  discussion of the initial conditions, such that the reader can easily perceive what has changed throughout the experiment and whether such changes might "modulate" the final results. I would expect that the time interval between filling the tanks and the start of the experiment could have exerted some effect in changing the initial conditions amongst different sites. But assuming that the 6 tanks were filled at the same time at each site, it is important to explain the reason why only 1 of the tanks/treatments changed.

Line 625-629: I don't understand what you mean here.

Line 638: I wouldn't say that it was "opposite" since DIP also decreased, only more abruptly compared top NOx.

Line 642: .... a dry Saharan dust deposition event was simulated...

Line 643-645: is it relevant to refer/repeat this? I would rather take this chance to

argument that your results appear to confirm previous notions that wet dust deposition is a more efficient source of bioavailable nutrients compared to dry dust deposition. . .

Line 645: Furthermore, based on previous studies reporting the . . . During the atmospheric transport of dust particles. . .

Line 650-654: this could be better explained. . .

Line 657: I would discuss first the drastic differences that you have found amongst the 3 sites before start comparing your observations with previous studies. The first sentence does not seem related to the sentence where you refer the study from Guieu and Ridame, 2020).

Line 659: I think you can safely say that the fertilization effects were much higher at ION and FAST compared to Guieu and Ridame, 2020.

Line 662: . . . with the largest NOx decrease observed in our study, which occurred at station FAST.

Line 668: . . . that, based in the analysis of several aerosol addition studies, Synechococcus had generally weak responses to aerosol addition. . .

Line 673: do you mean that Synechococcus was the group that increased the most in ION and FAST, compared to pico- and nano-eukaryotes?

Line 682: this is not clear: at the end of the experiment or following dust addition?

Line 689-692: It is not clear whether you are referring to clear evidences from your study or to info previously published. And what do you mean by "bacterial variability" in this context?

Line 685-718: I find this entire part very confusing and difficult to follow. . . please be more straight-to-the-point in what you think might be the explanation(s) for the lack of autotrophic response vs. positive heterotrophophic response to dust. And why was this the case for TYR but not FAST and ION. . .?

Line 725: a similar enhancement?

Line 748: what do you mean by an "abiotic dust experiment"?

Lines 757-758: This sounds contradictory with your previous sentence. If I understood correctly, you are saying that (A) while ocean warming and acidification do not seem to play any role on the amount of nutrient release from wet dust deposition in all three sites, (B) there are however differences amongst FAST, TYR and ION regarding the biological response to dust-born nutrients under ambient vs. warmer/acidified conditions. If this is the case, please explain it more clearly in the ms. And please justify/illustrate why you argue (A).

Line 760: why do you think you had larger variability amongst the duplicates at ION and TYR, but not at FAST? Was it due to methodological issues/limitations and/or to natural causes?

Line 763: I think you should discuss why you had enhanced biological stocks and metabolic rates at FAST under warm/acid conditions compared to ambient conditions before you start discussing why the differences in the DIP dynamics are harder to explain. . .

Line 763-771: this part is not clear. First you say that DIP was fully consumed until the end of the experiment at all 3 sites under future conditions, in contrast to ambient conditions. Immediately following, you seem to contradict yourself by saying that DIP dynamics at ION was similar between D and G treatments after all. . .. So, was there a difference in the DIP dynamics at all three sites or not?

Line 770: so you are saying that the dust-driven DIP was consumed much faster at TYR and FAST under future conditions compared to the present?

Line 774: throughout the ms. you often refer to "in preparation, this issue". If it is still in preparation than it is not "this issue". . . only after being submitted.

Line 786-787: are you referring to previous studies now? Please improve the transition

from your data to previous studies and vice-versa throughout the ms.

Line 795-797: Why is this evidenced by the lack of changes in the meso-zooplankton abundance? How exactly are the two related?

Line 810: why "excess production" instead of "enhanced production"?

Line 834: do you mean "under initial conditions"?

FIGURES

Fig. 1 – I would include lat/long in the location figure. The labels of most of the figures could be enlarged. Table 1 – It is not clear when exactly you have introduced the dust. Also, regarding the "related manuscripts": do you mean "in preparation" or "submitted"? Not sure you can/should quote studies that haven't been submitted yet.

---

## Referee Comment (RC2) · Anonymous Referee #2 · 14 Dec 2020

Dear authors,

Although I am far from an expert on the topic that you present in your manuscript, I decided to have a critical look at your manuscript since it turned out to be very difficult to find colleagues willing and able to assess your work. Also, the first reviewer did an excellent job by writing a very detailed comment on your manuscript. Frankly, I was impressed by the quality of your manuscript and I only have a few minor suggestions and questions, for which I refer to the annotated manuscript.

[Figure]

Please also note the supplement to this comment:
https://bg.copernicus.org/preprints/bg-2020-202/bg-2020-202-RC2-supplement.pdf

**Supplement:**

[revised manuscript text omitted]

0.68 m

1.09 m

LEDs

Air or air+CO$_2$ flush

Heater

PAR sensor

Sampling tube

Propeller

Sediment trap

Fig. 2.

[Figure]

 Fig. 3.

[Figure]

Fig. 4.

[Figure]

Fig. 5.

[Figure]

Fig. 6.

[Figure]

Fig. 7.

[Figure]

Fig. 8.

[Figure]

Fig. 9.

[Figure]

 Fig. 10.

---

## Author Comment (AC1) · 12 Feb 2021

**Anonymous Reviewer #2**

We would like to thank Anonymous Referee #2 for her/his comments and suggestions on our manuscript. We agree with most comments and modified/updated the manuscript accordingly. Below is a point-by-point reply, our answers appear in blue.

The project aimed at extensively studying and parameterizing the chain of processes occurring in the Mediterranean Sea after atmospheric deposition, especially of Saharan dust, and to put them in perspective of on-going environmental changes (Guieu et al., 2020).

> What else was deposited?
>
> This was removed since the objective of the cruise was to study the impact of all types of atmospheric particles.

both under present environmental conditions and following a realistic climate change scenario for 2100 (ca. +3 °C and -0.3 pH units; IPCC, 2013)

> This is probably a conservative scenario; see: Flecha, S., F. F. Pérez, J. García-Lafuente, S. Sammartino, A. F. Ríos, and I. E. Huertas (2015), Trends of pH decrease in the Mediterranean Sea through high frequency observational data: indication of ocean acidification in the basin, Scientific reports, 5, 16770-16770.
>
> This manuscript does not provide projections for the end of the century. Kapsenberg et al. (2017) provided a lower estimate of current pH decrease in the Mediterranean Sea on a longer time series. Anyway, we preferred considering a IPCC global scenario for the 2100 projection, which is actually close to the most pessimistic one as IPCC reports a -0.06 to -0.32 pH unit decrease depending on the socio-economic scenario considered (RCP2.6 to RCP 8.5).

The tanks are made of high-density polyethylene (HDPE) and are trace-metal free in order to avoid contaminations.

> How did you establish this?
>
> Because all the material used is made of HDPE, and was cleaned following the same protocols used for trace-metal free studies.

The tanks were filled by means of a large peristaltic pump (Verder© VF40 with EPDM hose, flow of 1200 L h-1) collecting seawater below the base of the boat (depth of ~ 5 m), used to supply continuously surface seawater to a series of instruments during the entire campaign.

> Were you always sailing? I could imagine that contamination would occur easily if you were not?
>
> The boat was "at station" while filling the tanks. This is a very normal way to work at sea. For example, all CTDs used during the cruise, including the Trace Metal Free Titanium Rosette equipped with GoFlo bottles, was deployed at station. The sampling at 5 m with this device does not indicate any type of contamination (Bressac et al., in prep.).

The particle size distribution showed that 99% of particles had a size smaller than 0.1 μm, and that particles were mostly made of quartz (40%), calcite (30%) and clay (25%; Desboeufs et al., 2014)

> This seems VERY fine grained to me; did you grind it? Before, you mentioned <20μm? Material this small will never settle (at least as individual particles....).
>
> Absolutely, thanks for pointing that up. We made a mistake that has been corrected: "particles had a size smaller than 1 μm."

The experiment at stations TYR and ION lasted 72 h (3 days) whereas the last experiment at station FAST was extended to four days.

> Why 3 days and why 4 days?
>
> This was added: This relatively short duration of the experiments was constrained by the time available between stations and the time needed to properly clean the tanks between the experiments, following the protocol described by Bressac and Guieu (2013). As a larger time window was possible at the end of the cruise, the experiment at FAST was extended to four days.

The identification of species was performed by automatic comparison with the library data set EcoTaxa (https://ecotaxa.obs-vlfr.fr/, last access: 17/04/2020) and then all validated and corrected by a human operator.

> i.e. you or one of your co-authors?

Modified to: "The identification of species was performed by automatic classification with a reference dataset in EcoTaxa (https://ecotaxa.obs-vlfr.fr/, last access: 17/04/2020) and then all validated and corrected manually.".

Overall, the differences between the warmed treatment (G) and the other tanks were +3, +3.2 and +3.6 °C at TYR, ION and FAST, respectively
Within 0.6 of each other; that is amazing!
Thank you!

However, at all three stations, initial concentrations of NOx (14, 18 and 59 nmol L-1 at TYR, ION and FAST, respectively; Table 2) were lower that the ones reported by Manca et al. (2004).
Replace that by than.
Sentence has been removed following suggestions from Reviewer #1.

Furthermore, as already mentioned, based on pigment analyses (HPLC), the sum of Fucoxanthin and Peridinin (representative of diatoms and dinoflagellates, respectively) represented only ~10% of the total chlorophyll a biomass at all stations. As biomass of both heterotrophic nanoflagellates and prokaryotes followed a west to east gradient (FAST > TYR > ION), ratio of autotrophic vs heterotrophic biomass appeared clearly in favor of the heterotrophic compartment at stations TYR and FAST (ratio of 0.6) while a value above the metabolic balance was estimated at ION (ratio of 1.3).
before you mentioned a lower number: <5%
Indeed, we clarified in the Results section: "At all three stations, the proportion of pigments representative of larger species (i.e. Fucoxanthin and Peridinin; diatoms and dinoflagellates respectively; Ras et al., 2008) were very small (< 5% for each pigment)".
Corrected to "the ratio"

NOx levels moderately decreased over the course of our experiments due to biological uptake
Reverse order
Modified

The opposite feature was observed for the DIP released by dust that rapidly decreased during our experiments
Reverse order
The sentence has been modified following Reviewer #1 suggestion: "While NOx levels decreased moderately over the course of our experiments due to biological uptake, more abrupt decreases were observed for DIP released by dust, reaching values close to the ones observed in the controls, except at station FAST where concentrations were still above ambient levels at the end of the experiment.".

Regarding the intensity of simulated wet deposition event, the 10 g m-2 deposition event considered here represents a high but realistic scenario, as several studies reported even higher short wet deposition events in this area of the Mediterranean Sea (Bonnet and Guieu, 2006; Loÿe-Pilot and Martin, 1996; Ternon et al., 2010).
Modified as suggested to: "The intensity of this simulated wet deposition event (i.e. 10 g m-2) represents a high but realistic scenario, as several studies reported even higher short wet deposition events in this area of the Mediterranean Sea (Bonnet and Guieu, 2006; Loÿe-Pilot and Martin, 1996; Ternon et al., 2010)".

This was especially true at station ION where no clear response to nutrient enrichment was observed for nano-eukaryotes throughout the experiment. However, it must be stressed that our experiments were performed over a relatively short period (3 to 4 days).
so why did you choose so (relatively) short experiment durations?
This is now explained in the Material and Methods section.

Feliu et al. (2020, this issue) have shown that the mesozooplankton assemblage at TYR was clearly impacted by a dust event that took place nine days before sampling at that station (François Dulac, Pers. Com. 2019) and evidenced by dust export in in situ deployed sediment traps (Bressac et al., in preparation, this issue).
it would be very interesting to study the particle-size distributions of the dust in this event sampled in the water column!
The sentence was modified as we actually have evidence directly from dustborn elements in the water column. Modified to: "Regarding the first hypothesis, Feliú et al. (2020) have shown that the mesozooplankton assemblage at TYR was clearly impacted by a dust event that took place nine days

before sampling at that station as evidenced as evidence from particulate inventory of lithogenic proxies (Al, Fe) in the water column (Bressac et al., in preparation).".

DIP levels decreased to reach similar levels than in control tanks at the end of this experiment (Fig. 6)
        Add as
        This section has been significantly modified. See revised version.

This results appears surprising as large impacts of warming and acidification have been observed,
        Results to Result
        Modified.

This is further evidenced by the absence of differences detected over the relatively short time duration of our experiment on meso-zooplankton abundance and carbon export efficiency (Gazeau et al., in preparation, this issue).
        Reverse order
        Modified.

At FAST, similar to what was observed at station ION, all phytoplanktonic groups were positively impacted by warming and acidification
        Reverse order
        Modified.

Also, in contrast to station ION, the abundance of heterotrophic prokaryotes in the warmer and acidified treatment reached a maximum after two days of incubations and then strongly decreased to reach levels observed in the control treatment.
        Reverse order
        Modified.

We fully acknowledge that the duration of our experiments was certainly too short to carefully assess the proportion of newly formed organic matter
        See addition in material and Methods.

---

## Author Comment (AC2) · 12 Feb 2021

We would like to thank Anonymous Referee #1 for her/his comments and suggestions on our manuscript. We also would like to apologize for the long time taken to revise this manuscript. We wanted to wait for the second review to make changes on our manuscript. That second review came several months after the first one (in December) and in the meantime, we have been very involved in the writing of other manuscripts from this special issue. We agree with most comments and modified/updated the manuscript accordingly. We especially acknowledge that the discussion was not well organized and focused on our results and their implications. We have tried our best to modify all sections from the discussion to make it clearer. Below is a point-by-point reply, our answers appear in blue. We will upload a "clean" version and a version with tracked changes, in order for you to visualize easily all the changes that have been made thanks to your hard work on our manuscript.

**General comments**: I consider the contribution by Gazeau et al., to be an extremely interesting and relevant study attempting at addressing important questions regarding the role of atmospheric dust deposition as an alternative source of bioavailable nutrients for ocean productivity in the warmer and more acidified ocean projected for the future.
The design of the experiment is excellent and very well-conceived. I particularly like the fact that it was run at 3 distinct environmental settings along a W-E gradient, providing an enormous potential for exploring changes/differences related to spatial/regional variability along the Mediterranean Sea.

This important study highlights how complex and diversified can be the responses of distinct biological groups to environmental variability, in this case, dust-born nutrient addition and ocean warming and acidification. Results from this work stress the urgent need for monitoring modern marine ecosystems and their spatiotemporal relationship to environmental parameters in order to better understand how they are/will respond to climate change. The study submitted by Gazeau et al., clearly has the potential to provide a robust discussion on these important and timely topics.

However, in its present form, the manuscript is not ready yet for publication, whereas a major revision and structure reorganization is recommended. Overall, I find that the paper lacks focus, and the interpretation of the results could benefit from some more maturation. The discussion, in particular, is hard to follow, bringing difficulties in getting a clear overview on how the different abiotic and biological parameters varied throughout time and space, and how the several hypotheses proposed for the observed trends are entangled/linked. The ideas could be better organized and entangled: whether you decide to compare each variable along different treatments/sites, or you decide to compare different sites in relation to all the variables (i.e. you provide a "general and integrated picture" for each site and there compare the three), whatever logic you use, please stick with it along the entire ms. for consistency. Overall, the main structure of the discussion could be organized in terms of the main differences abiotic and biotic observed along W-E gradient, and how such gradients in the initial/original in situ conditions are likely to modulate/be modulated by dust deposition and ocean warming and acidification.

Given the larger number of parameters and biological groups, for which you used different sub-sampling and analytical approaches, the set-up and procedure should be described more clearly, maybe adding an experiment schematic timeline. The manuscript could also benefit from a review in terms of written English.

**Specific comments:**

Section 4.1 The discussion would greatly benefit from being shortened and written in a more focused manner, instead of often repeating the results and providing too many unnecessary details that prevent the reader for getting the "general picture".
You should (a) provide a clear and focused global picture of the initial conditions along the FAST-TYR-ION transect, while (b) directly addressing potential causes/explanations for the most important/relevant patterns and observations described in the results.
We have tried our best to simplify this section and accommodate the referee's comments.

Section 4.2 I understand that the authors are discussing the strengths and limitations of their experiment, but I am not sure if they need an entire section for this, and especially not before discussing the results. What matters the most is the "story" they can tell from their spatiotemporal experiment, and what kind of insights on the effects of dust deposition and ocean warming/acidification they can extract from this story.

The section has been significantly shortened as suggested. However, we strongly believe that such a section is very important as these experimental systems have been used in the past and will be used in the future and, in our opinion, it is of the utmost importance to provide a critical assessment on those (what ameliorations were made, what still needs to be fixed etc…).

Section 4.3 Overall, this section could also benefit of more focus. The arguments should be better entangled while comparing a) different treatments, b) different sites, and c) with previous studies. Some parts are very confusing and difficult to follow. It is not easy for the reader to reach the end of this section having a clear picture on which groups benefited the most from dust, where and why, and how that translated into the whole ocean trophic chain and export of organic matter. Although this is the center-theme of the paper, it is not written in a very clear/straightforward manner. The authors should be able to discuss the difference between dust deposition along a W-E gradient along which the abiotic conditions vary. Potential impacts for the biological carbon pump could also be discussed.

Section 4.4 This section has similar issues than the previous sections, concerning its somewhat confusing and poorly organized writing/discussion. Furthermore, I also find it very similar to what is presented in the results. Following a clear and to-the-point presentation of the most striking differences between D and G treatments, the authors should take the chance to really discuss/reflect on the meaning of such differences in the light of ongoing ocean warming and acidification. The section revolves too much about details and differences amongst different sites/treatments but does not provide an actual "big picture" of the experiment nor a reflection on what the latter means.

Conclusion: I like this chapter because it is written following a more systematic, logic and to-the-point order. The discussion chapters would greatly benefit from being rewritten in a similar manner. In order to do that I would suggest the following approach:

- present a clear discussion of the initial conditions: you need to provide a clear abiotic and biotic spatial picture of your experiment. The environmental background is often not even mentioned. This is a pity since you have data for comparing three distinct environmental settings along a W-E gradient. A good discussion on the differences amongst different stations following dust/warming/acidification should take into account the environmental differences in their initial conditions.

- present a clear discussion on the effects of dust compared to the control, taking into account the differences in the initial conditions amongst the three sites. Differences between duplicates in each station and the potential causes for them should be referred after. Take the chance to discuss important questions such as: will increasing dust deposition increase ocean productivity, CO2 fixation and CO2 sequestration? If so, which groups are likely to contribute the most and where? Are dust effects likely to influence the entire marine trophic chain or not? Etc.

- present a clear discussion on the effects of dust in the projected future ocean compared to the modern ocean. Are the differences in the initial conditions a relevant factor for the observed differences and/or similarities? Can you say anything about ocean productivity and the biological pump of the future based on your observations? Will dust help to counterbalance the nutrient-depletion from ocean warming? What can you say about the effects of acidification on calcifying groups and does that matter? Etc.

Both sections have been significantly reorganized, and we do hope they provide now a much better view on our results and their implications.

**Abstract**
Line 26: caused by climate-driven enhanced stratification
Modified

Line 29: the potential impact (. . .) was investigated
Modified

Line 36: maybe "analysis" instead of "processes"?
Not modified, repetition of "analysis"

Line 38: impacts of dust seeding with and without addressing the effects of climate change
New sentence: "Here, we present the general setup of the experiments and the impacts of dust seeding with and without addressing the effects of environmental changes on nutrients and biological stocks."

Line 44: I would provide more information regarding such differences and regarding the conditions at the three studied locations. I would include more detailed information on which "biological stocks" you refer to, and which approach/method you used to explore the different biological groups.

We do not believe that going that deep into such information is necessary in our abstract (which is already very long…)

**Introduction**

Line 59: Please specify whether this Chl-a concentration refers to the surface, DCM or an average for the entire photic zone

New sentence: "The Mediterranean Sea is a typical example of these LNLC regions and exhibits surface chlorophyll a concentrations ...".

Line 65: can you explain/frame what are dust emissions from anthropogenic origin?

The sentence was rephrased: "Atmospheric deposition originates both from natural (mainly Saharan dust) and anthropogenic sources (e.g. Bergametti et al., 1989; Desboeufs et al., 2018). Dust deposition, mostly in the form of pulsed inputs, is mainly associated with wet deposition (Loÿe-Pilot and Martin, 1996).".

Line 76: how did these authors justify this lack of agreement?

As detailed in Guieu and Ridame, 2021, "dust events are often associated with strong wind events. Such wind, can be responsible for mixing of the waters, and could bring nutrients from below to the SML. In addition, Claustre et al. (2002) have shown that absorption and scattering by mineral dust particles suspended in surface waters after a deposition event likely produce a bias in SeaWiFS-derived Chla, resulting in an overestimation of the biomass, small but significant in such oligotrophic environment. It is thus very difficult, if not impossible, to disentangle both effects using only satellite images.

Line 62-76: I would break this paragraph in two.

Agreed. We have actually divided this section in 3 paragraphs as follows:

Paragraph 1: "Atmospheric deposition is well recognized as a significant source of micro- and macro-nutrients for surface waters of the global ocean … These regions are characterized by a low availability of macronutrients (N, P) and/or metal micronutrients (e.g. Fe) that can severely limit or co-limit phytoplankton growth during large periods of year."

Paragraph 2: "The Mediterranean Sea is a perfect example of these LNLC regions … In this region, the most important events reported in the 2010 decade amounted to ~22 g m-2 (Bonnet and Guieu, 2006; Guieu et al., 2010b)."

Paragraph 3: Atmospheric deposition provides new nutrients to the surface waters … although no clear correlation between dust and ocean color could be evidenced from long series of satellite observations in that part of the basin."

Line 78: in that part of the basin

Modified

Line 78-80: present a higher impact as a source of bioavailable fertilizing nutrients compared to dry deposition, as confirmed on. . .

Modified, see below.

Line 79-80: which parameters and processes were those?

Modified to: "Experimental approaches have shown that wet dust deposition events in the Northwestern Mediterranean Sea (the dominant deposition mode in that basin) present a higher impact as a source of bioavailable fertilizing nutrients compared to dry deposition. Indeed, wet deposition provides both new N and P while dry deposition supplies only P and does not allow to stimulate the

autotrophic community (except diazotrophs; Ridame et al., 2014), resulting in no increase in Chla concentration and primary production (Guieu et al., 2014a)."

Line 81: using both micro- and mesocosms
Modified

Line 83: while also modifying. . .
Modified

Line 84: In addition, besides. . . also modified the. . .
Modified

Line 87: can you explain what you mean with "budgets"?
Added: "carbon budget"

Line 89: . . . heterotrophic biological behavior in these oligotrophic waters
Modified

Line 94: how can dust aggregate and ballast OM that is dissolved?
Dust particles can favor the formation of TEPs (the formation of TEPs allows converting DOC into POC. That way, aggregates formed by particles/DOM will sink, the lithogenic particles act thus as a ballast for DOM.

Line 101: . . .nutrient cycling in the open ocean is being and will continue to be perturbed in the next decades, very likely to result in regionally variable impacts. . .
Modified

Line 104: . . . due to thermal stratification-linked reduction of nutrient supply. . . (. . .) to which an increasing role of atmospheric dust deposition might contribute to compensate as an alternative source. . ..
Sentence too long and too complicated. Suggestion: "Overall, LNLC areas are expected to expand in the future (Irwin and Oliver, 2009; Polovina et al., 2008) due to a thermal stratification related reduction of nutrient supply from sub-surface waters (Behrenfeld et al., 2006). As such, the role of atmospheric deposition might increase to compensate as an alternative source of new nutrients to surface waters.

Line 106: Whether or not plankton communities will respond differently to dust deposition under more acid and stratified conditions due to ongoing ocean warming and acidification, both globally and regionally in the Mediterranean Sea (refs.) is still largely unknown.
Alternate suggestion: "Whether or not plankton communities will respond differently to dust deposition considering the ongoing warming and acidification of the global ocean (IPCC, 2019), both processes also evidenced in the Mediterranean Sea (e.g. Kapsenberg et al., 2017; The Mermex group, 2011), is still largely unknown.".

Line 111: this sentence sounds strange; I would never expect that severe nutrient limitation and enhanced warming would ever lead to increase PP. . . or did you mean heterotrophic production?
Modified to: "As under severe nutrient limitation, warming has no effect on primary productivity (Marañón et al., 2018), it will most likely further push the balance towards net heterotrophy in oligotrophic areas.".

Line 119: what about calcifying plankton, which have an important goal for both the organic and inorganic carbon pumps?
We did not specifically look at coccolithophores during the study of Maugendre et al. (2015). Since no particular attention was also given to this group during the present study, we prefer not to introduce this aspect here. We showed in Oviedo, A.M., et al., Coccolithophore community response to increasing pCO2 in Mediterranean oligotrophic waters, Estuarine, Coastal and Shelf Science (2016), http://dx.doi.org/10.1016/j.ecss.2015.12.007, that OA exerts only a minor influence on this group in the Mediterranean Sea.

Line 144: this "and/or" is not very clear. . ..

Agreed. No need to repeat under present and future…, already mentioned in the previous sentence. Modified to "In this manuscript, we will present the general setup of the experiments and the evolution of nutrient and biological stocks.":

Line 145-147: do you mean other papers which were also submitted to this BG issue? "companion paper" does not sound well. . .

Modified to: "Several other manuscripts, related to these experiments and submitted or published in this special issue, focus on plankton metabolism (primary production, heterotrophic prokaryote production) and carbon export (Gazeau et al., 2021), on the microbial food web (Dinasquet et al., 2021), on nitrogen fixation (Ridame et al., in preparation) and on the release of insoluble elements (Fe, Al, REE, Th, Pa) from dust (Roy-Barman et al., 2020).

**2. Material and Methods**

2.1. General setup

Line 150: Six experimental tanks were installed in a T-controlled container, allowing to finely control the irradiance spectrum and intensity to fully reproduce a future scenario of enhanced ocean acidification and warming conditions.

Not modified. We believe the initial sentence was ok.

Line 153: were trace-metal free? Or ARE trace-metal free?

Modified: "are"

Line 180: how long did it take between sub-sampling for initial conditions and dust seeding? Can you trust that the conditions inside the tanks did not change during this interval (nutrient-consumption, grazing, OM degradation, etc)?

We understand that the experimental schedule was not clear enough, although some information were provided in Table 1. As such, we added a small schematic timeline as proposed by the reviewer on top of Table 1. Furthermore, we added this information in the following sentence : "While filling the tanks, this surface seawater was sampled for the measurements of selected parameters (sampling time = t-12h before dust seeding, see Table 1).".

Line 188: "for many parameters" sounds a bit vague..

Modified to "most parameters" and we refer to Table 1 for a clear view on what were samples for at each sampling point.

Line 205: I would only show the parameters that are used in the results/discussion of this paper.

We respectfully disagree, we believe it is important in this "overview" paper to show all parameters/processes for which samples were taken during the experiments.

Line 205: are 3 days enough? What were your criteria to define the time-interval for the experiments? Which biological groups were you focusing on? It would be interesting to monitor the ecological succession of distinct plankton groups following the addition of dust.

We agree that 3 days are a bit short to follow and monitor ecological succession. The main purpose was to observe any change in stocks and fluxes shortly after a dust deposition. In particular, the changes in biomass but also the rapid evolution of that biomass according to different phytoplankton groups could be observed. As mentioned in the revised version of the manuscript: "This relatively short duration of the experiments was constrained by the time available between stations and the time needed to properly clean the tanks between the experiments, following the protocol described by Bressac and Guieu (2013). As a larger time window was possible at the end of the cruise, the experiment at FAST was extended to four days."

We also added at the end of the introduction: "In this manuscript, we will present the general setup of the experiments and the evolution of nutrient and biological stocks (heterotrophic and autotrophic prokaryotes, photosynthetic eukaryotes as well as micro- and meso-zooplankton).".

Line 210: what do you mean by filtered "online"?

Modified as: "For some parameters (e.g. micro- and macro-nutrients), sampled seawater was directly filtered at the exit of the sampling tubes connected to each tank on membrane filter capsules (gravity filtration with Sartobran© 300; 0.2 μm). "

**2.2. Analytical methods**
2.2.2. Nutrients

Line 239: again the expression "filtered online".

Modified to: "Seawater samples for dissolved nutrients were filtered directly at the exit of the sampling tubes connected to each tank (Sartobran© 300; 0.2 μm), …"

2.2.4. Flow cytometry

Line 269: did you get any to details on the taxonomy of autotrophic nanoplankton?

Details on the taxonomy of autotrophic nanoplankton will be available in a manuscript that will be submitted in the coming days (referred to in the text as Dinasquet et al., 2021), as these authors did 16S rRNA and 18S rRNA sequencing, allowing to describe prokaryotic and eukaryotic communities. However, it would have been too much to detail taxonomy in this overview paper.

2.2.5. Micro-phytoplankton and -heterotrophs

Line 284: If the filling of the tanks took 2h, and if you have sampled such surface seawater while filling the tanks. . . why do you call this sampling time "t-12h"? The temporal sequence of the (1) filling, (2) filled-tanks, (3) dust seeding, (4) sub-sampling moments, and (5) which parameters you sub-sampled and when, is not fully clear to me. .

We refer to Table 1 for a clear understanding of the schedule. We do hope this is clearer now.

2.2.6. Mesozooplankton

Line 290: why was FAST longer? When exactly was the dust added to the tanks? T-?

Again, Table 1 provides this information.

2.3. Computation

What do you mean by "Computation"?

We modified to "Data analyses" and separated the section in 2 sub-sections.Alternative?
2.3.1. Nutrients inputs from dust
2.3.2. Autotrophic and heterotrophic biomass

Line 303: The maximum percentage of dust-born dissolved N and P. . .

Modified

Line 311: How many times did you subsample for nutrients over those 6h following dust seeding?

This is clarified now: "Based on maximal concentrations observed in the D and G tanks after seeding (two discrete sampling within 6 h following dust seeding, t1h and t6h), …"

Line 312: not sure I understand the way you determine the CONC dust. . .

We do not understand why it is not clear as we wrote "CONCdust corresponds to the maximum input of each nutrient, if 100% of its total concentration in the dust analog dissolves (as estimated based on dust chemical composition; Desboeufs et al., 2014; see above)."
But we modified to: "For each nutrient, CONCdust is the maximum addition, corresponding to a 100% dissolution of its total concentration in the dust analog (as estimated based on dust chemical composition; Desboeufs et al., 2014; see above).".

Line 325: . . . unites was done . . .

Modified

Line 331: what about Chl-a?

Bulk chlorophyll a does not allow discriminating taxonomic or size fractionated groups. We used the diagnostic pigments to have a better proxy of diatoms and dinoflagellates that we assumed to represent the majority of the micro-phytoplankton size class.

**3. Results**
**3.1. Initial conditions**

Line 335: the part: "when pumping seawater for the experiments" sounds repetition of the first sentence. Just directly start describing what were the initial conditions.
> Modified to: "Initial conditions of various measured parameters at the three sampling stations while filling the tanks (t-12h before seeding) are shown in Table 2. pHT and total alkalinity concentrations followed a west to east increasing gradient (8.03, 8.04 and 8.07; 2443, 2529 and 2627 µmol kg-1 at FAST, TYR and ION, respectively)."

Line 365: maybe this section should be titled "Experimental conditions at t0"
> We refer here to the experimental conditions not only at t0 but also during the experiments. We propose to rename this section as: "Conditions of irradiance, temperature and pH during the experiments"

Line 366: in the control tanks
> Modified to: "Irradiance levels during the experiments in the control tanks (C1, C2) …"

Lines 388-391: this part is not fully clear. Please specify to which tanks/treatments you added 13C and dust, and how was the sequence (which did you add first?)
> Modified to: "At all three stations, the addition of 13C-bicarbonate to all tanks before t0 led to an increase of total alkalinity between 6 and 11 µmol kg-1 at t0. Dust addition, performed right after t0 in tanks D and G, led to a AT decrease in these tanks between 8 and 16 µmol kg-1 at t24h with no apparent effects of warming and acidification."

Line 425: At TYR, while concentrations remained stable in control tanks, . . .
> Modified

Line 437: . . . temporal dynamics showed very different patterns amongst the three studied stations.
> Modified

Line 449: I would replace "at the exception of" by "with the exception of" throughout the ms.
> Modified

**4. Discussion**

**4.1. Initial conditions**

Line 499: typical stratified
> Modified to: "Overall, the three experiments were conducted with surface seawater collected during oligotrophic conditions typical of the open Mediterranean Sea at this period of the year (late spring).".

Line 500: please specify which period of the year you are referring to.
> See above

Line 501: . . . were even lower than the ones. . .. during spring
> Modified

Lines 514 and 515: please replace "whether. . .. or" by "both ". . . and"
> Modified

Lines 498-516: Maybe you don't need to provide such a long description given that your nutrient values are not directly comparable to those from the referred studies. It might be enough if you simply argue that in spite of not being able to directly compare them, your results suggest highly oligotrophic conditions at the start of the

experiment, in line with previous studies in the region and neighbor areas, and as typically expected for the open Mediterranean sea during spring. Basically, what you eventually say in lines 516-517.

This section has been significantly reduced as suggested by the reviewer.

Line 520: what do you mean by "enrichment experiments"? you mean experiments dealing with enrichment in DIP, nitrates, etc? And combination of what? Please explain.

Modified to: " "

Lines 520-527: it is not clear whether you are referring to experiments that you did or whether you are referring to studies by other authors (refs?). If I understood correctly, you are saying that productivity in the initial conditions were marked by N- and P limitation but not by dFe. . .?

See above. Modified to: "In contrast to N and P, initial concentrations of dissolved Fe in the sampled seawater, ranging from 1.5 nmol L-1 at TYR to 2.5 nmol L-1 at ION (Roy-Barman et al., 2020), were unlikely limiting for biological activity as previously shown in the Mediterranean Sea under stratified conditions (Bonnet et al., 2005; Ridame et al., 2014)."

Line 528: Total concentrations of Chl-a . . .. were in line with low Chl-a levels found. . ...both driven from remote sensing satellite images . . . and from in situ measurements provided in a database from. . ..

Modified, we just used "obtained" instead of "driven".

Line 531: . . . low Chl-a concentrations around. . .

Modified

Lines 533-538: this part sounds more like a description/repetition of the results rather than its actual discussion.

We tried to reformulate this sentence.

Line 540: . . . a fingerprint of LNLC areas in general, and of surface Mediterranean waters during spring in particular.

Modified sentence: "is a fingerprint of LNLC areas in general, and of surface Mediterranean waters in late spring and summer (Siokou-Frangou et al., 2010)."

Line 554: you should only cite works "submitted" or refer to them as "unpublished data" or "person. communication"

You are correct.

**4.2. Experimental assessment**

This section could benefit from being shortened and much more focused. As it is, the arguments are not easy to follow (often presented in "circles") making it hard to get a clear picture of what were in fact the initial conditions in the studied sites. For more details, see the comments below.

The section has been significantly shortened as suggested. However, we strongly believe that such a section is very important as these experimental systems have been used in the past and will be used in the future and, in our opinion, it is of the utmost importance to provide a critical assessment on those (what ameliorations were made, what still needs to be fixed etc...).

Line 568: Not sure if the title fits the content of this section; I would maybe call it "Environmental conditions during the experiment" or something.

The title was modified to: "4.2. Critical assessment of the experimental system and methodology".

Line 569: . . have been successfully validated in previous studies. . . to investigate the biological effects from the input of. . .. and resulting export of organic matter. . . under pre-defined close-to-abiotic conditions,

Not fully modified. These experiments were conducted as mentioned in close-to-abiotic conditions, as such the objective was not to test for "biological" effects. We believe the sentence is correct: "The experimental tanks used in this study have been successfully validated in previous studies designed to investigate the inputs of macro- and micro-nutrients (e.g. NOx, DIP, DFe) and the export of organic

matter, under close-to-abiotic conditions (seawater filtration onto 0.2 μm) following simulated wet dust events using the same analog as used in our study (Bressac and Guieu, 2013; Louis et al., 2017a, 2018).".

Line 574: which ones are "these"? please clarify.
Modified: "Louis et al. (2017a, 2018) further investigated these impacts under lowered pH conditions, although no control of atmospheric pCO2 was performed resulting in a rapid increase of pH levels in the acidified filtered seawater due to CO2 outgassing (from ~7.4 to ~7.7 in six days).".

Lines 574-575: . . . no control of atmospheric CO2 was, however, performed, resulting in a rapid increase of the pH levels in the acidified filtered seawater due to CO2 outgassing. . .
Modified, see above.

Line 576: In order to avoid this, we improved. . .. reactors). This allowed us to significantly reduce CO2 outgassing while maintaining pH levels. . ..
Modified: "This allowed to significantly reduce CO2 outgassing and maintain pH levels close to experimental targets.".

Line 580: Still, as illustrated in Fig. 5, the regulation of atmospheric CO2 was. . . compared to G1, possibly due to a potential. . ..
Modified: "Still, as illustrated in Fig. 5, the regulation of atmospheric CO2 was consistently more efficient in tank G2 compared to G1, resulting in a small discrepancy in terms of pH (highest difference of 0.04 pH units between the two G tanks at FAST), possibly due to a potential leak or a longer flushing time above tank G1.".

Line 583: I know what you mean here, but this should be said in a different way and properly supported by the relevant results.
Modified: "Nevertheless, as no systematic differences in terms of biological response were observed between these two tanks, we believe that these small differences in terms of regulated pH had no consequences on the obtained results.".

Line 588-592: too descriptive. Please be more concise and directly address the potential causes for the observed discrepancies.
If the lids are the same, why would the intensity vary? And what do you mean by "PARs sensors sensitivity"? Are you referring to enhanced levels of turbidity when you refer to "amount of particles"? And why only for treatments G?
Modified as suggested.

Line 597: you had the same type of unforeseen variability in terms of temperature? Only for G, or to all the treatments? This part should be better explained and discussed to what extent it might lead to misleading conclusions.
Modified to: "Continuous measurements in the tanks showed that temperature was not spatially homogeneous, leading to significant differences among replicates. This was especially the case for warmed tanks (treatment G) for which a maximal average difference over the experimental period of 0.7 °C was observed during the FAST experiment. As for the other controlled parameters discussed above, these discrepancies did not systematically lead to observable differences in the investigated stocks and processes between duplicates (except at TYR, see below).".

Line 599-600: ". . . but the new results are not yet available" (I am not sure what you want to state in the last sentence. . ..)
This sentence has been removed. We wanted to state that, after the PEACETIME cruise, we have developed a new container with a very fine temperature regulation, but we believe this can be removed.

Line 601: this is already well established at this point, there is no need to repeat this.
Removed

Line 605-612: Some of these sentences are too long. I am also not sure if this "introspection" about the criteria/available conditions for the design of your experiment is really relevant here. I would focus on providing a good discussion on the conditions prevailing during the experiment, how much they mimicked the settings that you wanted to mimic, how well the experiment conditions were in line with previous similar studies, and what is

it that is new. You can refer to the main difficulties and challenges during its implementation, but always having in mind whether the latter allowed you or not to realistically (enough?) simulate/test your hypothesis. You should stick to discuss the quality of your experiment based on your observations/results. Anything other than that creates dispersion and makes the reading more difficult.

This paragraph has been significantly reduced.

613-616: I understand this! However, this highlights the importance of providing a clear, to-the-point description discussion of the initial conditions, such that the reader can easily perceive what has changed throughout the experiment and whether such changes might "modulate" the final results. I would expect that the time interval between filling the tanks and the start of the experiment could have exerted some effect in changing the initial conditions amongst different sites. But assuming that the 6 tanks were filled at the same time at each site, it is important to explain the reason why only 1 of the tanks/treatments changed.

Unfortunately, we cannot provide a sound explanation on why one of the tanks behaved differently. This is the updated paragraph: "Nevertheless, we have to note that important discrepancies were detected regarding autotrophic stocks and processes (Gazeau et al., 2021) for tanks of the warmed and acidified treatment at station TYR. The reasons behind these differences are not fully understood but we strongly suspect that heterotrophic nano-flagellates, feeding mainly on prokaryotic picoplankton (Sherr and Sherr, 1994), exerted a strong top-down control on this group in tank G1 in which HNF abundance sharply increased during the experiment. All in all, while the methodology used in this study allowed to successfully evaluate the impacts of dust addition under both present and future environmental conditions at two of our three tested waters, these discrepancies at station TYR prevent us from drawing any strong conclusion on the effect of dust addition on the dynamics of the community under future environmental conditions at that station.".

Line 625-629: I don't understand what you mean here.

This paragraph has been simplified.

**4.3. Impact of dust addition**

Overall, this section could also benefit from more focus. The arguments should be better entangled while comparing a) different treatments, b) different sites, and c) with previous studies. Some parts are very confusing and difficult to follow. It is not easy for the reader to reach the end of this section having a clear picture on which groups benefited the most from dust, where and why, and how that translated into the whole ocean trophic chain and export of organic matter. Although this is the center-theme of the paper, it is not written in a very clear/straightforward manner. The authors should be able to discuss the difference between dust deposition along a W-E gradient along which the abiotic conditions vary. Potential impacts for the biological carbon pump could also be discussed.

This section has been significantly modified in order to accommodate the referee's comments.

Line 638: I wouldn't say that it was "opposite" since DIP also decreased, only more abruptly compared to NOx.

Modified.

Line 642: . . .. a dry Saharan dust deposition event was simulated. . .

Modified.

Line 643-645: is it relevant to refer/repeat this? I would rather take this chance to argue that your results appear to confirm previous notions that wet dust deposition is a more efficient source of bioavailable nutrients compared to dry dust deposition. . .

Modified.

Line 645: Furthermore, based on previous studies reporting the . . .. During the atmospheric transport of dust particles. . .

Modified.

Line 650-654: this could be better explained. . .

This paragraph has been moved to the start of the section. We do hope it is clearer.

Line 657: I would first discuss the drastic differences that you have found amongst the 3 sites before starting comparing your observations with previous studies. The first sentence does not seem related to the sentence where you refer to the study from Guieu and Ridame (2020).

> Modified as suggested.

Line 659: I think you can safely say that the fertilization effects were much higher at ION and FAST compared to Guieu and Ridame, 2020.

> This was added: "In contrast to what was observed at TYR, fertilization of primary producers was observed at stations ION and FAST under present conditions with overall relative changes much higher than from previous studies compiled by Guieu and Ridame (2020).".

Line 662: . . . with the largest NOx decrease observed in our study, which occurred at station FAST.

> Modified to: "The largest increase in chlorophyll a concentrations at station FAST is coherent with the largest NOx decrease observed in our study, which occurred at this station.".

Line 668: . . . that, based on the analysis of several aerosol addition studies, Synechococcus had generally weak responses to aerosol addition. . .

> Modified.

Line 673: do you mean that Synechococcus was the group that increased the most in ION and FAST, compared to pico- and nano-eukaryotes?

> Yes, we reformulated this whole paragraph as follows: "Although, in some cases, Synechococcus appeared stimulated by dust addition (Herut et al., 2005; Lagaria et al., 2017; Paytan et al., 2009), Guieu et al. (2014b) showed that, based on the analysis of several aerosols addition studies, this group had generally weak responses to aerosol addition in contrast to nano- and micro-phytoplankton, suggesting that aerosol deposition may lead to an increase in larger size class phytoplankton. Yet, at stations ION and FAST, the increase in Synechococcus abundance in dust-amended tanks was the highest relative to those of pico- and nano-eukaryotes.".

Line 682: this is not clear: at the end of the experiment or following dust addition?

> Modified to: "Although this was not observed based on pigment analyses, the sharp decline in nano-eukaryote abundances in dust-amended tanks at the end of the FAST experiment, further suggests that this group, reacting quickly to nutrient enrichment was progressively grazed and/or outcompeted by larger phytoplankton species.".

Line 689-692: It is not clear whether you are referring to clear evidence from your study or to info previously published. And what do you mean by "bacterial variability" in this context?

> Modified to: "As not only phytoplankton but also heterotrophic bacteria are limited by inorganic nutrients, mainly DIP, in oligotrophic systems (Obernosterer et al., 2003; Wambeke et al., 2002), many recent studies have shown significant increase in heterotrophic bacterial abundance, respiration and/or production following dust deposition in these areas (Lekunberri et al., 2010; Pitta et al., 2017; Pulido-Villena et al., 2008, 2014; Romero et al., 2011)."

Line 685-718: I find this entire part very confusing and difficult to follow. . . please be more straight-to-the-point in what you think might be the explanation(s) for the lack of autotrophic response vs. positive heterotrophic response to dust. And why was this the case for TYR but not FAST and ION. . .?

> Modified as suggested.

Line 725: a similar enhancement?

> Modified.

4.4. Impact of warming and acidification

This section has similar issues than the previous sections, concerning its somewhat confusing and poorly organized writing/discussion. Furthermore, I also find it very similar to what is presented in the results. Following a clear and to-the-point presentation of the most striking differences between D and G treatments, the authors should take the chance to really discuss/reflect on the meaning of such differences in the light of

ongoing ocean warming and acidification. The section revolves too much about details and differences amongst different sites/treatments but does not provide an actual "big picture" of the experiment nor a reflection on what the latter means.

This section has been significantly modified in order to accommodate the referee's comments.

Line 748: what do you mean by an "abiotic dust experiment"?
Modified to: "Louis et al. (2018) have already shown from an experiment performed under close-to-abiotic conditions (seawater filtration onto 0.2 µm) that even an extreme ocean acidification scenario…".

Lines 757-758: This sounds contradictory with your previous sentence. If I understood correctly, you are saying that (A) while ocean warming and acidification do not seem to play any role on the amount of nutrient release from wet dust deposition in all three sites, (B) there are however differences amongst FAST, TYR and ION regarding the biological response to dust-born nutrients under ambient vs. warmer/acidified conditions.
If this is the case, please explain it more clearly in the ms. And please justify/illustrate why you argue (A).
Rephrased as: "As no differences were observed for NOx and DIP concentrations as measured within few hours following dust addition under present and future environmental conditions, our results agree with these previous findings and further highlights the absence of direct effect of ocean warming (+3 °C) on the release of nutrients from atmospheric particles.
In contrast, following these similar nutrient releases, different nutrient consumption dynamics were observed between ambient and warmed/acidified tanks. These differences were substantially dependent on the considered nutrient and investigated station.".

Line 760: why do you think you had larger variability amongst the duplicates at ION and TYR, but not at FAST? Was it due to methodological issues/limitations and/or to natural causes?
Modified to: "Regarding NOx, no impacts of warming and acidification could be observed at stations TYR and ION due to low net decreasing rates compared to the large initial stock released from dust. In contrast, at the most productive station FAST, as a consequence of strongly enhanced biological stocks (see thereafter) and metabolic rates (Gazeau et al., 2021). larger NOx consumption rates were shown under future environmental conditions.".

Line 763: I think you should discuss why you had enhanced biological stocks and metabolic rates at FAST under warm/acid conditions compared to ambient conditions before you start discussing why the differences in the DIP dynamics are harder to explain. . .
We respectfully disagree as we would like to keep the same logic as in the previous section, meaning discussing nutrient dynamics and then biological stocks.

Line 763-771: this part is not clear. First you say that DIP was fully consumed until the end of the experiment at all 3 sites under future conditions, in contrast to ambient conditions. Immediately following, you seem to contradict yourself by saying that DIP dynamics at ION was similar between D and G treatments after all. . ..
So, was there a difference in the DIP dynamics at all three sites or not?
Yes, we tried to rephrase: "A clear feature of our experiments is that, in contrast to present day pH and temperature conditions, all the stock of DIP released from dust was consumed at the end of the three experiments under future conditions. That being said, the rate of decrease under future environmental conditions differed depending on the station.'.
*Explanation: Yes, indeed DIP was fully consumed at the end of the 3 experiments in G tanks, which was not the case at FAST in D tanks. However, the speed at which DIP was consumed differed between stations, i.e. it took the same time than for D tanks (basically the whole experiment) at ION while it was all consumed within 24 h at TYR and FAST.*

Line 770: So you are saying that the dust-driven DIP was consumed much faster at TYR and FAST under future conditions compared to the present?
Rephrased: "While DIP dynamics were relatively similar between tanks maintained under present and future environmental conditions at ION, warming and acidification induced a faster decrease of DIP at TYR and FAST, with a full consumption of the released DIP within 24 h.".

Line 774: throughout the ms. you often refer to "in preparation, this issue". If it is still in preparation then it is not "this issue". . . only after being submitted.

Among the companion papers, Gazeau et al. (2021) and Dinasquet et al. (2021) will be submitted in the coming days and we prefer mentioning (2021). Only Ridame et al. (2021) is not ready yet for submission and we will refer to this manuscript in preparation as "unpublished results"

Line 786-787: are you referring to previous studies now? Please improve the transition from your data to previous studies and vice-versa throughout the ms.

Modified as: "These positive effects of warming and acidification on the abundance of phytoplankton cells, especially for small species, as observed at ION and FAST are coherent with previously published studies. Indeed, although very contrasted results have been shown on the effect of ocean acidification on small autotrophic species (e.g. Dutkiewicz et al., 2015), there is increasing evidence that small phytoplankton species will be favored in a warmer ocean (e.g. Chen et al., 2014; Daufresne et al., 2009; Morán et al., 2010). As mentioned earlier, our experimental protocol was not conceived to discriminate temperature from pH effects, however results concur with those of Maugendre et al. (2015) which further suggested temperature over elevated CO2 as the main driver of increased picophytoplankton abundance. ".

Line 795-797: Why is this evidenced by the lack of changes in the meso-zooplankton abundance? How exactly are the two related?

This sentence has been removed.

Line 810: why "excess production" instead of "enhanced production"?

Modified.

Line 834: do you mean "under initial conditions"?

Yes, modified.

**FIGURES**

Fig. 1 – I would include lat/long in the location figure. The labels of most of the figures could be enlarged. Table 1 – It is not clear when exactly you have introduced the dust. Also, regarding the "related manuscripts": do you mean "in preparation" or "submitted"? Not sure you can/should quote studies that haven't been submitted yet

Fig. 1 has been modified by the map below and labels from all figures have been enlarged as suggested.

---

## Editor Decision (ED1)

[revised manuscript text omitted]

4% (see Gazeau et al., 2021).  valve at the base of  tanks was   the remaining water inside the tanks ( 165-180 L;  172.5 L and  150 L)

through a  PVC sieve . The organisms retained  were gently removed using a washing bottle filled with filtered seawater (0.2 µm), and transferred directly in a 250 mL bottle.

4%.  using a

ZooSCAN (Hydroptic©; Gorsky et al., 2010) at the PIQv-platform of EMBRC-France.

automatic classification with a reference dataset in

EcoTaxa (https://ecotaxa.obs-vlfr.fr/, last access: 17/04/2020) and  validat

.

**2.3.1. Nutrient inputs from dust**

The maximum percentage of dust-born dissolved N and P was     Based on maximal concentrations observed in  D and G   t1h and t6h

$$\%_{dissolution} = \frac{CONC_{max} - CONC_{init}}{CONC_{dust}} \cdot 100 \tag{1}$$

where $CONC_{init}$ is the concentration of the corresponding nutrient in each tank before seeding (t0), $CONC_{max}$ corresponds to the concentration of the corresponding nutrient in each tank when nutrient concentration was at a maximum  the first 6 h after seeding  , and $CONC_{dust}$ is the maximum  a 100% dissolution  dust analog ( based on dust ; Desboeufs et al., 2014; ).

**2.3.2. Autotrophic and heterotrophic biomass**

 micro-phytoplankton , as a first approximation, autotrophic biomass was  as the sum of  *Synechococcus,* pico- and nano-eukaryotes ( based on flow cytometry)   carbon units was  250 fg C cell$^{-1}$ (Kana and Glibert, 1987),  Verity et al. (1992;  
[revised manuscript text omitted]

[Figure]

Fig. 1.

[Figure]

Fig. 2.

[Figure]

Fig. 3.

[Figure]

[Figure]

Fig. 4.

[Figure]

Fig. 5.

[Figure]

Fig. 6.

[Figure]

Fig. 7.

[Figure]

Fig. 8.

[Figure]

Fig. 9.

[Figure]

Fig. 10.

---

## Author Response (AR2)

**Villefranche-sur-mer,July 7th 2021**

Dear Christine,

Many thanks for your hard work on our manuscript. The vast majority of suggestions you made were implemented. The few ones that were not followed are answered directly in the PDF below.

We hope to have satisfactorily accommodated your comments and that it can be now accepted for publication.

Thank you for considering our manuscript. We look forward hearing from you in due course.

Sincerely yours,

Frédéric Gazeau

[Figure]

**Laboratoire d'Océanographie de
Villefranche, LOV
Sorbonne Université-CNRS
Chemin du Lazaret
06230 Villefranche-sur-Mer, France**

[revised manuscript text omitted]